# ONLINE MAP INFERENCE AND LEARNING FOR NONSYMMETRIC DETERMINANTAL POINT PROCESSES

## ABSTRACT

In this paper, we introduce the online and streaming MAP inference and learning problems for Non-symmetric Determinantal Point Processes (NDPPs) where data points arrive in an arbitrary order and the algorithms are constrained to use a single-pass over the data as well as sub-linear memory. The online setting has an additional requirement of maintaining a valid solution at any point in time. For solving these new problems, we propose algorithms with theoretical guarantees, evaluate them on several real-world datasets, and show that they give comparable performance to state-of-the-art offline algorithms that store the entire data in memory and take multiple passes over it.

## 1 INTRODUCTION

Determinantal Point Processes (DPPs) were first introduced in the context of quantum mechanics (Macchi, 1975) and have subsequently been extensively studied with applications in several areas of pure and applied mathematics like graph theory, combinatorics, random matrix theory (Hough et al., 2006; Borodin, 2009), and randomized numerical linear algebra (Derezinski & Mahoney, 2021). Discrete DPPs have gained widespread adoption in machine learning following the seminal work of Kulesza & Taskar (2012) and there has been a recent explosion of interest in DPPs in the machine learning community. For instance, some of the very recent uses of DPPs include automation of deep neural network design (Nguyen et al., 2021), deep generative models (Chen & Ahmed, 2021), document and video summarization (Perez-Beltrachini & Lapata, 2021), image processing (Launay et al., 2021), and learning in games (Perez-Nieves et al., 2021).

A DPP is a probability distribution over subsets of items and is characterized by some kernel matrix such that the probability of sampling any particular subset is proportional to the determinant of the submatrix corresponding to that subset in the kernel. Until very recently, most prior work on DPPs focused on the setting where the kernel matrix is symmetric. Due to this constraint, DPPs can only model negative correlations between items. Recent work has shown that allowing the kernel matrix to be nonsymmetric can greatly increase the expressive power of DPPs and allows them to model *compatible* sets of items (Gartrell et al., 2019; Brunel, 2018). To differentiate this line of work from prior literature on symmetric DPPs, the term Nonsymmetric DPPs (NDPPs) has often been used. Modeling positive correlations can be useful in many practical scenarios. For instance, an E-commerce company trying to build a product recommendation system would want the system to increase the probability of suggesting a router if a customer adds a modem to a shopping cart.

State-of-the-art algorithms for learning and inference on NDPPs (Gartrell et al., 2021) require storing the full data in memory and take multiple passes over the complete dataset. Therefore, these algorithms take too much memory to be useful for large scale data, where the size of the entire dataset can be much larger than the random-access memory available. These algorithms are also not practical in settings where data is generated on the fly, for example, in E-commerce applications where new items are added to the store over time, and more importantly, added to the carts of users instantaneously.

This work makes the following contributions:

**Streaming and Online Inference:** We formulate streaming and online versions of maximum a posteriori (MAP) inference on fixed-size NDPPs and provide algorithms for solving these problems. In the streaming setting, data points arrive in an arbitrary order and the algorithms are constrained to

use a single-pass over the data as well as sub-linear memory (i.e. memory that is substantially smaller than the size of the data stream). The online setting we consider has an additional restriction that we need to maintain a valid solution at every time step. For both these settings, we provide algorithms which have comparable or even better solution quality than the offline greedy algorithm while taking only a single pass over the data and using a fraction of the memory used by the offline algorithm.

**Online Learning:** We introduce the online learning problem for NDPPs and provide an algorithm which solves this problem using a single-pass over the data and memory that is constant in $m$, the number of baskets in the training data (or equivalently the length of the stream). In comparison, the offline learning algorithm takes a large number of passes over the entire data and uses memory linear in $m$. Strikingly, our online learning algorithm shows comparable performance (log-likelihood) to the state-of-the-art offline learning algorithm, while converging significantly faster in all cases (Figure 2). This is notable, since our algorithm uses only a single pass over the data, while using a tiny fraction of the memory.

Table 1: Summary of our online learning and MAP inference algorithms for NDPPs. We omit $O$ for simplicity. All algorithms use only a single-pass over the data.

| NDPP Problem | | | Update Time | Total Time | Space |
|---|---|---|---|---|---|
| **Inference** | **Streaming** | STREAM-PARTITION (Alg. 1) | $\mathcal{T}_{\det}(k,d)$ | $\mathcal{T}_{\det}(k,d)\cdot n$ | $k^2+d^2$ |
| | **Online** | ONLINE-LSS (Alg. 2) | $\mathcal{T}_{\det}(k,d)\cdot k\log^2(\Delta)$ | $\mathcal{T}_{\det}(k,d)\cdot(nk+k\log^2(\Delta))$ | $k^2+d^2+d\log_\alpha(\Delta)$ |
| | | ONLINE 2-NEIGH (Alg. 3) | $\mathcal{T}_{\det}(k,d)\cdot k^2\log^3(\Delta)$ | $\mathcal{T}_{\det}(k,d)\cdot(nk^2+k^2\log^3(\Delta))$ | $k^2+d^2+d\log_\alpha(\Delta)$ |
| | | ONLINE-GREEDY (Alg. 5) | $\mathcal{T}_{\det}(k,d)\cdot k$ | $\mathcal{T}_{\det}(k,d)\cdot nk$ | $k^2+d^2$ |
| **Online Learning** | | ONLINE-NDPP (Alg. 4) | $d^3 + d'^3$ | $md'^3 + md^3$ | $d'^2 + d^2$ |

## 2  RELATED WORK

Even in the case of (symmetric) DPPs, the study of online and streaming settings is in a nascent stage. In particular, Bhaskara et al. (2020) were the first to propose online algorithms for MAP inference of DPPs and Liu et al. (2021) were the first to give streaming algorithms for the maximum induced cardinality objective proposed by Gillenwater et al. (2018). However, no work has focused on either online or streaming MAP inference or online learning for Nonsymmetric DPPs.

A special subset of NDPPs called signed DPPs were the first class of NDPPs to be studied (Brunel et al., 2017). Gartrell et al. (2019) studied a more general class of NDPPs and provided learning and MAP Inference algorithms, and also showed that NDPPs have additional expressiveness over symmetric DPPs and can better model certain problems. This was improved by Gartrell et al. (2021) in which they provided a new decomposition which enabled linear time learning and MAP Inference for NDPPs. More recently, Anari & Vuong (2021) proposed the first algorithm with a $k^{O(k)}$ approximation factor for MAP Inference on NDPPs where $k$ is the number of items to be selected. These works are not amenable to the streaming nor online settings that are studied in our paper. In particular, they store all data in memory and use multiple passes over the data, among other issues. In this work, we formally introduce the streaming and online MAP inference and online learning problems for NDPPs and develop online algorithms for solving these problems. To the best of our knowledge, our work is the first to study NDPPs in the streaming and online settings, and develop algorithms for solving MAP inference and learning of NDPPs in these settings.

## 3  PRELIMINARIES

**Notation.**  Throughout the paper, we use uppercase bold letters ($\boldsymbol{A}$) to denote matrices and lowercase bold letters ($\boldsymbol{a}$) to denote vectors. Letters in normal font ($a$) will be used for scalars. For any positive integer $n$, we use $[n]$ to denote the set $\{1, 2, \ldots, n\}$. A matrix $\boldsymbol{M}$ is said to be skew-symmetric if $\boldsymbol{M} = -\boldsymbol{M}^\top$ where $\top$ is used to represent matrix transposition.

A DPP is a probability distribution on all subsets of $[n]$ characterized by a matrix $\boldsymbol{L} \in \mathbb{R}^{n \times n}$. The probability of sampling any subset $S \subseteq [n]$ i.e. $\Pr[S] \propto \det(\boldsymbol{L}_S)$ where $\boldsymbol{L}_S$ is the submatrix of $\boldsymbol{L}$

obtained by keeping only the rows and columns corresponding to indices in $S$. The normalization constant for this distribution can be computed efficiently since we know that $\sum_{S \subseteq [n]} \det(\boldsymbol{L}_S) = \det(\boldsymbol{L} + \boldsymbol{I}_n)$ (Kulesza & Taskar, 2012, Theorem 2.1). Therefore, $\Pr[S] = \frac{\det(\boldsymbol{L}_S)}{\det(\boldsymbol{L} + \boldsymbol{I}_n)}$. For the DPP corresponding to $\boldsymbol{L}$ to be a valid probability distribution, we need $\det(\boldsymbol{L}_S) \geq 0$ for all $S \subseteq [n]$ since $\Pr[S] \geq 0$ for all $S \subseteq [n]$. Matrices which satisfy this property are known as $P_0$-matrices (Fiedler & Pták, 1966). For any symmetric matrix $\boldsymbol{L}$, $\det(\boldsymbol{L}_S) \geq 0$ for all $S \subseteq [n]$ if and only if $\boldsymbol{L}$ is positive semi-definite (PSD) i.e. $\boldsymbol{x}^T \boldsymbol{L} \boldsymbol{x} \geq 0$ for all $\boldsymbol{x} \in \mathbb{R}^n$. Therefore, all symmetric matrices which correspond to valid DPPs are PSD. But there are $P_0$-matrices which are not necessarily symmetric (or even positive semi-definite). For example, $\boldsymbol{L} = \begin{bmatrix} 1 & 1 \\ -1 & 1 \end{bmatrix}$ is a nonsymmetric $P_0$ matrix.

Any matrix $\boldsymbol{L}$ can be uniquely written as the sum of a symmetric and skew-symmetric matrix: $\boldsymbol{L} = (\boldsymbol{L} + \boldsymbol{L}^\top)/2 + (\boldsymbol{L} - \boldsymbol{L}^\top)/2$. For the DPP characterized by $\boldsymbol{L}$, the symmetric part of the decomposition can be thought of as encoding negative correlations between items and the skew-symmetric part as encoding positive correlations. Gartrell et al. (2019) proposed a decomposition which covers the set of all nonsymmetric PSD matrices (a subset of $P_0$ matrices) which allowed them to provide a cubic time algorithm (in the ground set size) for NDPP learning. This decomposition is $\boldsymbol{L} = \boldsymbol{V}^\top \boldsymbol{V} + (\boldsymbol{B} \boldsymbol{C}^\top - \boldsymbol{C} \boldsymbol{B}^\top)$. Gartrell et al. (2021) provided more efficient (linear time) algorithms for learning and MAP inference using a new decomposition $\boldsymbol{L} = \boldsymbol{V}^\top \boldsymbol{V} + \boldsymbol{B}^\top \boldsymbol{C} \boldsymbol{B}$. Although both these decompositions only cover a subset of $P_0$ matrices, it turns out that they are quite useful for modeling real world instances and provide improved results when compared to (symmetric) DPPs.

For the decomposition $\boldsymbol{L} = \boldsymbol{V}^\top \boldsymbol{V} + \boldsymbol{B}^\top \boldsymbol{C} \boldsymbol{B}$, we have $\boldsymbol{V}, \boldsymbol{B} \in \mathbb{R}^{d \times n}, \boldsymbol{C} \in \mathbb{R}^{d \times d}$ and $\boldsymbol{C}$ is skew-symmetric. Here we can think of the items having having a latent low-dimensional representation $(\boldsymbol{v}_i, \boldsymbol{b}_i)$ where $\boldsymbol{v}_i, \boldsymbol{b}_i \in \mathbb{R}^d$. Intuitively, a low-dimensional representation (when compared to $n$) is sufficient for representing items because any particular item only interacts with a small number of other items in real-world datasets, as evidenced by the fact that the maximum basket size encountered in real-world data is much smaller than $n$.

## 4 STREAMING MAP INFERENCE

In this section, we formulate the streaming MAP inference problem for NDPPs and design an algorithm for this problem with guarantees on the solution quality, space, and time.

### 4.1 STREAMING MAP INFERENCE PROBLEM

We study the MAP Inference problem in low-rank NDPPs in the streaming setting where we see columns of a $2d \times n$ matrix in order (column-arrival model). Given some fixed skew-symmetric matrix $C \in \mathbb{R}^{d \times d}$, consider a stream of $2d$-dimensional vectors (which can be viewed as pairs of $d$-dimensional vectors) arriving in order:

$$(\boldsymbol{v}_1, \boldsymbol{b}_2), (\boldsymbol{v}_2, \boldsymbol{b}_2), \ldots, (\boldsymbol{v}_n, \boldsymbol{b}_n) \text{ where } \boldsymbol{v}_t, \boldsymbol{b}_t \in \mathbb{R}^d, \forall\, t \in [n]$$

The main goal in the streaming setting is to output the maximum likelihood subset $S \subseteq [n]$ of cardinality $k$ at the end of the stream assuming that $S$ is drawn from the NDPP characterized by $\boldsymbol{L} = \boldsymbol{V}^\top \boldsymbol{V} + \boldsymbol{B}^\top \boldsymbol{C} \boldsymbol{B}$ i.e.

$$S = \operatorname*{argmax}_{S \subseteq [n], |S| = k} \det(\boldsymbol{L}_S) = \operatorname*{argmax}_{S \subseteq [n], |S| = k} \det(\boldsymbol{V}_S^\top \boldsymbol{V}_S + \boldsymbol{B}_S^\top \boldsymbol{C} \boldsymbol{B}_S) \tag{1}$$

For any $S \subseteq [n], \boldsymbol{V}_S \in \mathbb{R}^{d \times |S|}$ is the matrix whose each column corresponds to $\{\boldsymbol{v}_i, i \in S\}$. Similarly, $\boldsymbol{B}_S \in \mathbb{R}^{d \times |S|}$ is the matrix whose columns correspond to $\{\boldsymbol{b}_i, i \in S\}$. In the case of symmetric DPPs, this maximization problem in the non-streaming setting corresponds to MAP Inference in cardinality constrained DPPs, also known as $k$-DPPs (Kulesza & Taskar, 2011).

Usually, designing a streaming algorithm can be viewed as a dynamic data-structure design problem. We want to maintain a data-structure with efficient time and small space over the entire stream. Therefore, the secondary goals are to minimize the following:

- Space: We consider word-RAM model, and use number of words[*] to measure it.

---

[*] In word-RAM, we usually assume each word is $O(\log n)$ bits.

---

**Algorithm 1** Streaming Partition Greedy MAP Inference for low-rank NDPPs

---

1: **Input:** Length of the stream $n$ and a stream of data points $\{(\boldsymbol{v}_1, \boldsymbol{b}_1), (\boldsymbol{v}_2, \boldsymbol{b}_2), \ldots, (\boldsymbol{v}_n, \boldsymbol{b}_n)\}$
2: **Output:** A solution set $S$ of cardinality $k$ at the end of the stream.
3: $S_0 \leftarrow \emptyset$, $s_0 \leftarrow \emptyset$
4: **while** new data $(\boldsymbol{v}_t, \boldsymbol{b}_t)$ arrives in stream at time $t$ **do**
5:      $i \leftarrow \lceil \frac{tk}{n} \rceil$
6:      **if** $f(S_{i-1} \cup \{t\}) > f(S_{i-1} \cup \{s_i\})$ **then**
7:         $s_i \leftarrow t$
8:      **if** $t$ is a multiple of $\frac{n}{k}$ **then**
9:         $S_i \leftarrow S_{i-1} \cup s_i$
10:         $s_i \leftarrow \emptyset$
11: **return** $S_k$

---

- Update time: time to update our data-structure whenever we see a new arriving data point.

- Total time: total time taken to process the stream.

**Definition 1.** *Given three matrices $\boldsymbol{V} \in \mathbb{R}^{d \times k}$, $\boldsymbol{B} \in \mathbb{R}^{d \times k}$ and $\boldsymbol{C} \in \mathbb{R}^{d \times d}$, let $\mathcal{T}_{\det}(k, d)$ denote the running time of computing*

$$\det(\boldsymbol{V}^\top \boldsymbol{V} + \boldsymbol{B}^\top \boldsymbol{C} \boldsymbol{B}).$$

*Note that $\mathcal{T}_{\det}(k, d) = 2\mathcal{T}_{\mathrm{mat}}(d, k, d) + \mathcal{T}_{\mathrm{mat}}(d, d, k) + \mathcal{T}_{\mathrm{mat}}(k, k, k)$ where $\mathcal{T}_{\mathrm{mat}}(a, b, c)$ is the time required to multiply two matrices of dimensions $a \times b$ and $b \times c$. We have the last $\mathcal{T}_{\mathrm{mat}}(k, k, k)$ term because computing the determinant of a $k \times k$ matrix can be done (essentially) in the same time as computing the product of two matrices of dimension $k \times k$ (Aho et al., 1974, Theorem 6.6).*

We will now describe a streaming algorithm for MAP inference in NDPPs, which we call the "Streaming Partition Greedy" algorithm.

### 4.2 STREAMING PARTITION GREEDY

**Outline of Algorithm 1**: Our algorithm picks the first element of the solution greedily from the first seen $\frac{n}{k}$ elements, the second element from the next sequence of $\frac{n}{k}$ elements and so on. As described in Algorithm 1, let us use $S_0, S_1, \ldots, S_k$ to denote the solution sets maintained by the algorithm, where $S_i$ represents the solution set of size $i$. In particular, we have that $S_i = S_{i-1} \cup \{s_i\}$ where $s_i = \arg\max_{j \in \mathcal{P}_i} f(S \cup \{j\})$ and $\mathcal{P}_i$ denotes the $i$'th partition of the data i.e. $\mathcal{P}_i := \{\frac{(i-1) \cdot n}{k} + 1, \frac{(i-1) \cdot n}{k} + 2, \ldots, \frac{i \cdot n}{k}\}$.

$$\underbrace{(\boldsymbol{v}_1, \boldsymbol{b}_1), (\boldsymbol{v}_2, \boldsymbol{b}_2), ..., (\boldsymbol{v}_{n/k}, \boldsymbol{b}_{n/k})}_{\mathcal{P}_1}, \ldots, \underbrace{(\boldsymbol{v}_{n-(n/k)+1}, \boldsymbol{b}_{n-(n/k)+1}), ..., (\boldsymbol{v}_n, \boldsymbol{b}_n)}_{\mathcal{P}_k} \quad (2)$$

**Theorem 2.** *For a random-order arrival stream, if $S$ is the solution output by Algorithm 1 at the end of the stream and $\sigma_{\min} > 1$ where $\sigma_{\min}$ and $\sigma_{\max}$ denote the smallest and largest singular values of $\boldsymbol{L}_S$ among all $S \subseteq [n]$ and $|S| \leq 2k$, then*

$$\mathbb{E}[\log \det(\boldsymbol{L}_S)] \geq \left( 1 - \frac{1}{\sigma_{\min}^{(1-\frac{1}{e}) \cdot (2 \log \sigma_{\max} - \log \sigma_{\min})}} \right) \log(\mathrm{OPT})$$

*where $\boldsymbol{L}_S = \boldsymbol{V}_S^\top \boldsymbol{V}_S + \boldsymbol{B}_S^\top \boldsymbol{C} \boldsymbol{B}_S$ and $\mathrm{OPT} = \max_{R \subseteq [n], |R|=k} \det(\boldsymbol{V}_R^\top \boldsymbol{V}_R + \boldsymbol{B}_R^\top \boldsymbol{C} \boldsymbol{B}_R)$.*

We defer the proofs of the above and the following theorem to Appendix A.

**Theorem 3.** *For any length-$n$ stream $(\boldsymbol{v}_1, \boldsymbol{b}_1), \ldots, (\boldsymbol{v}_n, \boldsymbol{b}_n)$ where $(\boldsymbol{v}_t, \boldsymbol{b}_t) \in \mathbb{R}^d \times \mathbb{R}^d \ \forall \ t \in [n]$, the worst-case update time of Algorithm 1 is $O(\mathcal{T}_{\det}(k, d))$ where $\mathcal{T}_{\det}(k, d)$ is the time taken to compute $f(S) = \det(\boldsymbol{V}_S^\top \boldsymbol{V} + \boldsymbol{B}_S^\top \boldsymbol{C} \boldsymbol{B})$ for $|S| = k$. The total time taken is $O(n \cdot \mathcal{T}_{\det}(k, d))$ and the space used at any time step is $O(k^2 + d^2)$.*

---

**Algorithm 2** ONLINE-LSS: Online MAP Inference for low-rank NDPPs with Stash.

---

1: **Input:** A stream of data points $\{(\boldsymbol{v}_1, \boldsymbol{b}_1), (\boldsymbol{v}_2, \boldsymbol{b}_2), \ldots, (\boldsymbol{v}_n, \boldsymbol{b}_n)\}$, and a constant $\alpha \geq 1$
2: **Output:** A solution set $S$ of cardinality $k$ at the end of the stream.
3: $S, T \leftarrow \emptyset$
4: **while** new data point $(\boldsymbol{v}_t, \boldsymbol{b}_t)$ arrives in stream at time $t$ **do**
5:      **if** $|S| < k$ and $f(S \cup \{t\}) \neq 0$ **then**
6:          $S \leftarrow S \cup \{t\}$
7:      **else**
8:          $i \leftarrow \arg\max_{j \in S} f(S \cup \{t\} \setminus \{j\})$
9:          **if** $f(S \cup \{t\} \setminus \{i\}) > \alpha \cdot f(S)$ **then**
10:             $S \leftarrow S \cup \{t\} \setminus \{i\}$
11:             $T \leftarrow T \cup \{i\}$
12:             **while** $\exists \, a \in S, \, b \in T : f(S \cup \{b\} \setminus \{a\}) > \alpha \cdot f(S)$ **do**
13:                 $S \leftarrow S \cup \{b\} \setminus \{a\}$
14:                 $T \leftarrow T \cup \{a\} \setminus \{b\}$
15: **return** $S$

---

# 5 ONLINE MAP INFERENCE FOR NDPPS

We now consider the online MAP inference problem for NDPPs, which is natural in settings where data is generated on the fly. In addition to the constraints of the streaming setting (Section 4.1), our online setting requires us to maintain a valid solution at every time step. In this section, we provide two algorithms for solving this problem.

## 5.1 ONLINE LOCAL SEARCH WITH A STASH

**Outline of Algorithm 2**: On a high-level, our algorithm is a generalization of the Online-LS algorithm for DPPs from Bhaskara et al. (2020). At each time step $t \in [n]$ (after $t \geq k$), our algorithm maintains a candidate solution subset of indices $S$ of cardinality $k$ from the data seen so far i.e. $S \subseteq [t]$ s.t. $|S| = k$ in a streaming fashion. Additionally, it also maintains two matrices $\boldsymbol{V}_S, \boldsymbol{B}_S \in \mathbb{R}^{d \times |S|}$ where the columns of $\boldsymbol{V}_S$ are $\{\boldsymbol{v}_i, i \in S\}$ and the columns of $\boldsymbol{B}_S$ are $\{\boldsymbol{b}_i, i \in S\}$. Whenever the algorithm sees a new data point $(\boldsymbol{v}_t, \boldsymbol{b}_t)$, it replaces an existing index from $S$ with the newly arrived index if doing so increases $f(S)$ at-least by a factor of $\alpha \geq 1$ where $\alpha$ is a parameter to be chosen (we can think of $\alpha$ being 2 for understanding the algorithm). Instead of just deleting the index replaced from $S$, it is stored in an auxiliary set $T$ called the "stash" (and also maintains corresponding matrices $\boldsymbol{V}_T, \boldsymbol{B}_T$), which the algorithm then uses to performs a local search over to find a locally optimal solution.

We now define a data-dependent parameter $\Delta$ which we will need to describe the time and space used by Algorithm 2.

**Definition 4.** *Let the first non-zero value of $f(S)$ with $|S| = k$ that can be achieved in the stream without any swaps be* $\mathrm{val}_{nz}$ *i.e. till $S$ reaches a size $k$, any index seen is added to $S$ if $f(S)$ remains non-zero even after adding it. Let us define* $\Delta := \frac{\mathrm{OPT}_k}{\mathrm{val}_{nz}}$ *where* $\mathrm{OPT}_k = \max_{S \subseteq [n], |S| = k} \det(\boldsymbol{V}_S^\top \boldsymbol{V}_S + \boldsymbol{B}_S^\top C \boldsymbol{B}_S)$.

**Theorem 5.** *For any length-$n$ stream $(\boldsymbol{v}_1, \boldsymbol{b}_1), \ldots, (\boldsymbol{v}_n, \boldsymbol{b}_n)$ where $(\boldsymbol{v}_t, \boldsymbol{b}_t) \in \mathbb{R}^d \times \mathbb{R}^d \, \forall \, t \in [n]$, the worst case update time of Algorithm 2 is $O(\mathcal{T}_{\det}(k, d) \cdot k \log^2(\Delta))$ where $\mathcal{T}_{\det}(k, d)$ is the time taken to compute $f(S) = \det(\boldsymbol{V}_S^\top \boldsymbol{V} + \boldsymbol{B}_S^\top C \boldsymbol{B})$ for $|S| = k$. Furthermore, the amortized update time is $O(\mathcal{T}_{\det}(k, d) \cdot (k + \frac{k \log^2(\Delta)}{n}))$ and the space used at any time step is at most $O(k^2 + d^2 + d \log_\alpha(\Delta))$.*

We defer the proof for the above theorem and all following theorems in this section to Appendix B.

## 5.2 ONLINE 2-NEIGHBORHOOD LOCAL SEARCH ALGORITHM WITH A STASH

Before we describe our algorithm, we will define a *neighborhood* of any solution, which will be useful for describing the local search part of our algorithm.

---

**Algorithm 3** ONLINE-2-NEIGHBOR: Local Search over 2-neighborhoods with Stash for Online MAP Inference of low-rank NDPPs.

---

1: **Input:** A stream of data points $\{(\boldsymbol{v}_1, \boldsymbol{b}_1), (\boldsymbol{v}_2, \boldsymbol{b}_2), \ldots, (\boldsymbol{v}_n, \boldsymbol{b}_n)\}$, and a constant $\alpha \geq 1$
2: **Output:** A solution set $S$ of cardinality $k$ at the end of the stream.
3: $S, T \leftarrow \emptyset$
4: **while** new data $(\boldsymbol{v}_t, \boldsymbol{b}_t)$ arrives in stream at time $t$ **do**
5:      **if** $|S| < k$ and $f(S \cup \{t\}) \neq 0$ **then**
6:          $S \leftarrow S \cup \{t\}$
7:      **else**
8:          $\{i, j\} \leftarrow \arg\max_{a,b \in S} (f(S \cup \{t\} \setminus \{a\}), f(S \cup \{t-1, t\} \setminus \{a, b\}))$
9:          $f_{\max} \leftarrow \max_{a,b \in S} (f(S \cup \{t\} \setminus \{a\}), f(S \cup \{t-1, t\} \setminus \{a, b\}))$
10:         **if** $f_{\max} > \alpha \cdot f(S)$ **then**
11:             **if** two items are chosen to be replaced: **then**
12:                $S \leftarrow S \cup \{t-1, t\} \setminus \{i, j\}$
13:                $T \leftarrow T \cup \{i, j\}$
14:             **else**
15:                $S \leftarrow S \cup \{t\} \setminus \{i\}$
16:                $T \leftarrow T \cup \{i\}$
17:             **while** $\exists\, a, b \in S,\ c, d \in T : f(S \cup \{c, d\} \setminus \{a, b\}) > \alpha \cdot f(S)$ **do**
18:                $S \leftarrow S \cup \{c, d\} \setminus \{a, b\}$
19:                $T \leftarrow T \cup \{a, b\} \setminus \{c, d\}$
20: **return** $S$

---

**Definition 6** ($\mathcal{N}_r(S, T)$). *For any natural number $r \geq 1$ and any sets $S, T$ we define the $r$-neighborhood of $S$ with respect to $T$*

$$\mathcal{N}_r(S, T) := \{S' \subseteq S \cup T \mid |S'| = |S| \text{ and } |S' \setminus S| \leq r\}$$

**Outline of Algorithm 3**: Similar to Algorithm 2, our new algorithm also maintains two subsets of indices $S$ and $T$, and corresponding data matrices $\boldsymbol{V}_S, \boldsymbol{B}_S, \boldsymbol{V}_T, \boldsymbol{B}_T$. Whenever the algorithm sees a new data-point $(\boldsymbol{v}_t, \boldsymbol{b}_t)$, it checks if the solution quality $f(S)$ can be improved by a factor of $\alpha$ by replacing any element in $S$ with the newly seen data-point. Additionally, it also checks if the solution quality can be made better by including both the points $(\boldsymbol{v}_t, \boldsymbol{b}_t)$ and the data-point $(\boldsymbol{v}_{t-1}, \boldsymbol{b}_{t-1})$. Further, the algorithm tries to improve the solution quality by performing a local search on $\mathcal{N}_2(S, T)$ i.e. the neighborhood of the candidate solution $S$ using the stash $T$ by replacing at most two elements of $S$. There might be interactions captured by *pairs* of items which are much stronger than single items in NDPPs (see example 5 from Anari & Vuong (2021)).

A full description of Algorithm 3 can be found in Appendix B.

**Theorem 7.** *For any length-$n$ stream $(\boldsymbol{v}_1, \boldsymbol{b}_1), \ldots, (\boldsymbol{v}_n, \boldsymbol{b}_n)$ where $(\boldsymbol{v}_t, \boldsymbol{b}_t) \in \mathbb{R}^d \times \mathbb{R}^d\ \forall\ t \in [n]$, the worst case update time of Algorithm 3 is $O(\mathcal{T}_{\det}(k, d) \cdot k^2 \log^3(\Delta))$ where $\mathcal{T}_{\det}(k, d)$ is the time taken to compute $f(S) = \det(\boldsymbol{V}_S^\top \boldsymbol{V} + \boldsymbol{B}_S^\top \boldsymbol{C} \boldsymbol{B})$ for $|S| = k$. The amortized update time is $O\left(\mathcal{T}_{\det}(k, d) \cdot \left(k^2 + \frac{k^2 \log^3(\Delta)}{n}\right)\right)$ and the space used at any time step is at most $O(k^2 + d^2 + d \log_\alpha(\Delta))$.*

## 6 ONLINE LEARNING

In this section, we will first formally introduce the online learning problem for NDPPs and then develop an online algorithm for this new setting with theoretical guarantees. To the best of our knowledge, this is the first work to study the online learning setting for NDPPs.

### 6.1 ONLINE SETTING

The online learning problem for NDPPs is formally defined as follows:

**Definition 8** (Online NDPP Learning). *Given a continuous stream of observed sets $\{S_1, \ldots, S_t, \ldots, \}$ of items from $[n]$, the online learning problem is to maintain an NDPP kernel*

---

**Algorithm 4** ONLINE-NDPP-LEARNING: Online learning for low-rank NDPPs
---
1: **Input:** Stream of sets $\{S_1, S_2, \ldots, S_t, \ldots\}$ arriving in an online fashion (in some arbitrary order), embedding size $d$ for NDPP model matrices
2: **Output:** low-rank NDPP kernel $\boldsymbol{L}^{(t)}$ at any time $t$ (and $\boldsymbol{V}^{(t)}, \boldsymbol{B}^{(t)}, \boldsymbol{C}^{(t)}$ if warranted)
3: Initialize matrices $\boldsymbol{V}, \boldsymbol{B}, \boldsymbol{C}$
4: **while** new set $S_t = \{a_1, a_2, \ldots\}$ arrives in stream **do**
5:     Update $\boldsymbol{V}_{S_t}, \boldsymbol{B}_{S_t}, \boldsymbol{C}$ using $\nabla_{\boldsymbol{V}_{S_t}} \psi_t, \nabla_{\boldsymbol{B}_{S_t}} \psi_t, \nabla_{\boldsymbol{C}} \psi_t$ (Eqs. 9, 10, 11) respectively.
6: **return** low-rank NDPP kernel $\boldsymbol{L}^{(t)}$ at time $t$

---

$\boldsymbol{L}_t = (\boldsymbol{V}^\top \boldsymbol{V} + \boldsymbol{B}^\top \boldsymbol{C} \boldsymbol{B})_t$ *for every time step $t$ that maximizes the regularized log-likelihood:*

$$\phi_t(\boldsymbol{V}, \boldsymbol{B}, \boldsymbol{C}) = \frac{1}{t} \sum_{i=1}^{t} \log \det \left( \boldsymbol{V}_{S_i}^\top \boldsymbol{V}_{S_i} + \boldsymbol{B}_{S_i}^\top \boldsymbol{C} \boldsymbol{B}_{S_i} \right) - \log \det \left( \boldsymbol{V}^\top \boldsymbol{V} + \boldsymbol{B}^\top \boldsymbol{C} \boldsymbol{B} + \boldsymbol{I} \right) - R(\boldsymbol{V}, \boldsymbol{B}) \quad (3)$$

*where $\boldsymbol{V}_{S_i}, \boldsymbol{B}_{S_i} \in \mathbb{R}^{d \times |S_i|}$ denote sub-matrices of $\boldsymbol{V}$ and $\boldsymbol{B}$ which are formed by the columns that correspond to the items in $S_i$, and the regularizer $R(\boldsymbol{V}, \boldsymbol{B}) = \alpha \sum_{i=1}^{n} \frac{1}{\mu_i} \|\boldsymbol{v}_i\|_2^2 + \beta \sum_{i=1}^{n} \frac{1}{\mu_i} \|\boldsymbol{b}_i\|_2^2$ where $\mu_i$ is the number of occurrences of the item $i$ in all the training data (subsets), $\alpha, \beta > 0$ are tunable hyperparameters.*

*An online learning algorithm for NDPPs needs to:*

- *Use space that is independent of the length of the stream seen so far.*
- *Use only a single sequential pass over the stream of subsets $\{S_i \subseteq [n] \mid i = 1 \text{ to } t\}$.*
- *Update the NDPP model upon arrival of the subset $S_t$ at time $t$, then discards it.*
- *Have a fast update time that is sub-linear in the number of unique items $n$.*

Our online setting is the natural setting to be studied for learning NDPPs in the real-world, as new subsets are generated continuously over time, for instance in an E-commerce application. In contrast, the state-of-the-art learning algorithm Gartrell et al. (2021) has the following limitations that make it fundamentally offline: First, it stores all training data in memory and so uses $O(td')$ space where $d' = \max_i |S_i|$ is the size of the largest subset in the stream $\{S_1, \ldots, S_t\}$. Second, it takes multiple passes over the data. Third, subsets are not processed as they arrive sequentially, instead that algorithm requires all data to be stored in memory, and updates a large group of points simultaneously over multiple rounds. Thus, incurring a large processing time that is not amenable to the online learning setting. Finally, it also incurs an update time of $O(pnd^2 + ptd'^3)$, which is too much for the online setting.

## 6.2 ONLINE ALGORITHM

We present our online learning approach for NDPPs in Algorithm 4. To update $(\boldsymbol{V}, \boldsymbol{B}, \boldsymbol{C})$ at every time step $t$, we use the gradient of the following objective function $\psi_t$, which is an approximation of the regularized log-likelihood $\phi_t(\boldsymbol{V}, \boldsymbol{B}, \boldsymbol{C})$. Let us define $Z(\boldsymbol{V}, \boldsymbol{B}, \boldsymbol{C}) := \log \det \left( \boldsymbol{V}^\top \boldsymbol{V} + \boldsymbol{B}^\top \boldsymbol{C} \boldsymbol{B} + \boldsymbol{I} \right)$. Our approximate objective

$$\psi_t(\boldsymbol{V}, \boldsymbol{B}, \boldsymbol{C}) = \log \det \left( \boldsymbol{V}_{S_t}^\top \boldsymbol{V}_{S_t} + \boldsymbol{B}_{S_t}^\top \boldsymbol{C} \boldsymbol{B}_{S_t} \right) - Z(\boldsymbol{V}_{S_t}, \boldsymbol{B}_{S_t}, \boldsymbol{C}) - R(\boldsymbol{V}_{S_t}, \boldsymbol{B}_{S_t}) \quad (4)$$

For every time step $t$, we only update $\boldsymbol{V}_{S_t}, \boldsymbol{B}_{S_t}, \boldsymbol{C}$ using the gradient of $\psi_t$ i.e. $\nabla_{\boldsymbol{V}_{S_t}, \boldsymbol{B}_{S_t}, \boldsymbol{C}} \psi_t$. We defer the gradient derivations to Appendix D.

We defer the proof of the following theorem to Appendix D.

**Theorem 9.** *Algorithm 4 uses $O(d'^2 + d^2)$ space [†]. Also, the update time for a single subset $S_t$ arriving at time $t$ in the stream is $O(d^3 + d'^3)$ where $d'$ is the size of the largest set in the stream.*

For every new subset $S_t$ arriving in the stream, Algorithm 4 updates only the columns of $\boldsymbol{V}$ and $\boldsymbol{B}$ corresponding to the elements in $S_t$, hence, $\boldsymbol{V}_{S_t}, \boldsymbol{B}_{S_t}$. It uses total space $O(nd + d'^2)$, which is

---

[†]Note that the $O(nd + d^2)$ required to store the matrices $\boldsymbol{V}, \boldsymbol{B}$, and $\boldsymbol{C}$ are omitted for simplicity (since these terms are common to all algorithms).

independent of the stream length (and is therefore sub-linear in it as well); takes a single pass over the stream; updates the NDPP model only using the subset $S_t$ and then discards $S_t$; has an $O(d^3 + d'^3)$ update time. This is in contrast to the offline learning algorithm that uses $O(md')$ space where $m$ is the length of the stream and $d'$ is the size of the largest set ($m \gg d, d', n$).

## 7 EXPERIMENTS

Our experiments are designed to evaluate the effectiveness of the proposed online inference and learning algorithms for NDPPs. Experiments are performed using a standard desktop computer (Quad-Core Intel Core i7, 16 GB RAM) using many real-world datasets. Details of the datasets can be found in Appendix E.

### 7.1 ONLINE INFERENCE

As a point of comparison, we also use the state-of-the-art offline greedy algorithm from Gartrell et al. (2021). This algorithm stores all data in memory and makes $k$ passes over the dataset and in each round, picks the data point which gives the maximum marginal gain in solution value. Online-Greedy replaces a point in the current solution set with the observed point if doing so increases the objective. See Algorithm 5 in Appendix B for more details.

We first learn all the matrices $B, C$, and $V$ by applying the offline learning algorithm. The learning algorithm takes as input a parameter $d$, which is the embedding size for $V, B, C$. We use $d = 10$ for all datasets other than Instacart, Customer Dashboards, Company Retail where $d = 50$ is used and Million Song, where $d = 100$ is used. We perform MAP inference by running our algorithms on the learned $B, C$, and $V$. For all the following results, we set $\alpha = 1.1$ and $k = 8$.

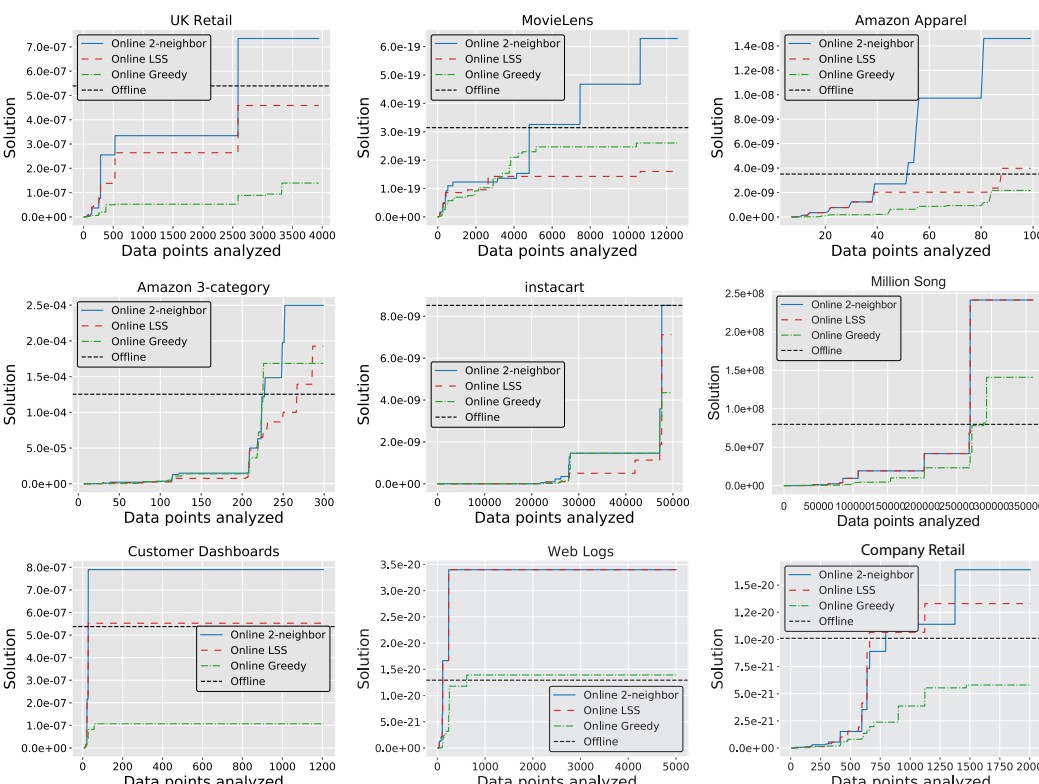

Figure 1: Solution quality i.e. objective function value as a function of the number of data points analyzed for all our online algorithms and also the offline greedy algorithm. All our online algorithms give comparable (or even better) performance to offline greedy using only a single pass and a small fraction of the memory.

MAP Inference results for a variety of different datasets are provided in Figure 1. Surprisingly, the solution quality of our online algorithms compare favorably with the offline greedy algorithm while using only a single-pass over the data, and a tiny fraction of memory. In most cases, Online-2-neighbor (Algorithm 3) performs better than Online-LSS (Algorithm 2) which in turn performs better than the online greedy algorithm (Algorithm 5). Strikingly, our online-2-neighbor algorithm performs even better than offline greedy in many cases. Furthermore, in some cases the other online algorithms also perform better than the offline greedy algorithm.

We also perform several experiments comparing the number of determinant computations (as a system-independent proxy for time) and the number of swaps (as a measure of solution consistency) of all our online algorithms. Results for determinant computations (Figure 3) and swaps (Figure 4) can be found in Appendix F. We summarize the main findings here. The number of determinant computations of Online-LSS is comparable to that of Online Greedy but the number of swaps performed is significantly smaller. Online-2-neighbor is the most time-consuming but superior performing algorithm in terms of solution quality.

We also investigate the performance of our algorithms under the random stream paradigm, where we consider a random permutation of some of the datasets used earlier. Results for the solution quality (Figure 5), number of determinant computations and swaps (Figure 6) can be found in Appendix F.3. In this setting, we see that Online-LSS and Online-2-neighbor have nearly identical performance and are always better than Online-Greedy in terms of solution quality and number of swaps.

We study the effect of varying $\alpha$ in Online-LSS (Algorithm 2) for various values of set sizes $k$ in Appendices F.4 and F.5. We notice that, in general, the solution quality, number of determinant computations, and the number of swaps increase as $\alpha$ decreases (Figure 7). We also see that as $k$ increases, the solution value decreases across all values of $\alpha$ (Figure 8). This is in accordance with our intuition that the probability of larger sets should be smaller.

## 7.2 ONLINE LEARNING RESULTS

Here, we compare our online learning algorithm for NDPPs with the state-of-the-art offline learning algorithm. In Figure 2, we compare our online learning algorithm to the offline algorithm using negative log-likelihood over time. In all cases, the offline learning algorithm uses significantly more time, usually taking between 6x to 15x more time to converge. Furthermore, the negative log-likelihood from our online learning algorithm decreases significantly faster compared to the offline.

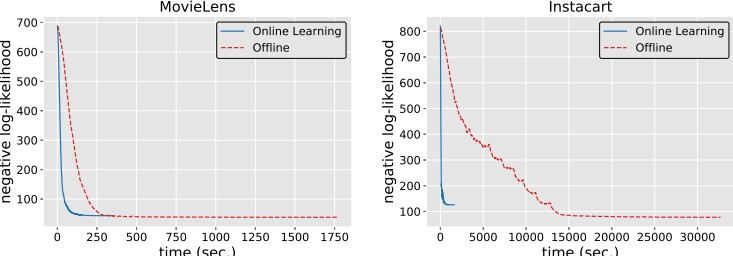

Figure 2: Results comparing our online NDPP learning algorithm to the offline learning algorithm. Strikingly, our online learning algorithm shows comparable performance, while converging significantly faster in all cases. Furthermore, it also uses only a single pass, while using a small fraction of the space. See text for discussion.

## 8 CONCLUSION

In this paper, we formulate and study the streaming and online MAP inference and learning problems for Nonsymmetric Determinantal Point Processes. To the best of our knowledge, this is the first work to study these problems in these practical settings. We design new algorithms for these problems, prove theoretical guarantees for them in terms of space required, time taken, and solution quality for our algorithms, and empirically show that they perform comparably or (sometimes) even better than state-of-the-art offline algorithms. We believe our work opens up completely new avenues for practical application of NDPPs and can lead to deployment of NDPPs in many more real-world settings.

## REPRODUCIBILITY STATEMENT

For all of our theoretical results, we have provided clear explanations of all the assumptions in the main text (Sections 4, 5, and 6) and have also provided complete proofs in the corresponding sections in the Appendix (Sections A, B, and D). For our experimental results, we have provided all details needed to reproduce our results (like the setup etc) in the main text (Section 7) and have also provided details of the datasets used in our experiments in Appendix E.

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

## A  STREAMING MAP INFERENCE DETAILS

**Theorem 2.** *For a random-order arrival stream, if $S$ is the solution output by Algorithm 1 at the end of the stream and $\sigma_{\min} > 1$ where $\sigma_{\min}$ and $\sigma_{\max}$ denote the smallest and largest singular values of $\boldsymbol{L}_S$ among all $S \subseteq [n]$ and $|S| \leq 2k$, then*

$$\mathbb{E}[\log \det(\boldsymbol{L}_S)] \geq \left(1 - \frac{1}{\sigma_{\min}^{(1-\frac{1}{e}) \cdot (2\log\sigma_{\max} - \log\sigma_{\min})}}\right) \log(\text{OPT})$$

*where $\boldsymbol{L}_S = \boldsymbol{V}_S^\top \boldsymbol{V}_S + \boldsymbol{B}_S^\top \boldsymbol{C} \boldsymbol{B}_S$ and $\text{OPT} = \max\limits_{R \subseteq [n],\ |R|=k} \det(\boldsymbol{V}_R^\top \boldsymbol{V}_R + \boldsymbol{B}_R^\top \boldsymbol{C} \boldsymbol{B}_R).$*

We will first give a high-level proof sketch for our this theorem.

*Proof sketch.* For a random-order arrival stream, the distribution of any set of consecutive $n/k$ elements is the same as the distribution of $n/k$ elements picked uniformly at random (without replacement) from $[n]$. If the objective function $f$ is submodular, then this algorithm has an approximation guarantee of $(1 - 2/e)$ by Mirzasoleiman et al. (2015). But neither $\det(\boldsymbol{L}_S)$ nor $\log\det(\boldsymbol{L}_S)$ are submodular. Instead, Gartrell et al. (2021)[Equation 45] showed that $\log\det(\boldsymbol{L}_S)$ is "close" to submodular when $\sigma_{\min} > 1$ where this closeness is measured using a parameter known as "submodularity ratio" (Bian et al., 2017). Using this parameter, we can prove a guarantee for our algorithm. ∎

We will now provide a more complete proof.

*Proof.* As described in Algorithm 1, we will use $S_0, S_1, \ldots, S_k$ to denote the solution sets maintained by the algorithm, where $S_i$ represents the solution set of size $i$. In particular, we have that $S_i = S_{i-1} \cup \{s_i\}$ where $s_i = \arg\max_{j \in B_i} f(S \cup \{j\})$ and $B_i$ denotes the $i$'th partition i.e. $B_i := \{\frac{(i-1)\cdot n}{k} + 1, \frac{(i-1)\cdot n}{k} + 2, \ldots, \frac{i \cdot n}{k}\}$.

For $i \in [k]$, let us use $X_i := [B_i \cap (S_* \setminus S_{i-1}) \neq \emptyset]$ to denote the event that there is at least one element of the optimal solution which has not already been picked by the algorithm in the batch $B_i$ and $\lambda_i := |S_* \setminus S_{i-1}|$. Then,

$$\begin{aligned}
\Pr[X_i] &= 1 - \Pr[X_i^c] \\
&= 1 - (1 - \frac{\lambda_i}{n})(1 - \frac{\lambda_i}{n-1}) \cdots (1 - \frac{\lambda_i}{n - \frac{n}{k} + 1}) \\
&\geq 1 - \left(1 - \frac{\lambda_i}{n}\right)^{\frac{n}{k}} \\
&\geq 1 - e^{-\frac{\lambda_i}{k}} \\
&\geq \frac{\lambda_i}{k} \cdot \left(1 - \frac{1}{e}\right)
\end{aligned}$$

Here we use the facts: $e^x \geq 1 + x$ for all $x \in \mathbb{R}$, $1 - e^{-\frac{\lambda}{k}}$ is concave as a function of $\lambda$, and $\lambda \in [0, k]$.

For any element $s \in [n]$ and set $S \subseteq [n]$, let us use $f(s \mid S) := f(S \cup \{s\}) - f(S)$ to denote the marginal gain in $f$ obtained by adding the element $s$ to the set $S$. For any round $i \in [k]$, we then have that $f(S_i) - f(S_{i-1}) = f(s_i \mid S_{i-1})$.

Note that

$$\mathbb{E}[f(s_i \mid S_{i-1}) \mid X_i] \geq \frac{\sum_{\omega \in \text{OPT} \setminus S_{i-1}} f(\omega \mid S_{i-1})}{|\text{OPT} \setminus S_{i-1}|}.$$

This happens due to the fact that conditioned on $X_i$, every element in $S_* \setminus S_{i-1}$ is equally likely to be present in $B_i$ and the algorithm picks $s_i$ such that $f(s_i \mid S_{i-1}) \geq f(s \mid S_{i-1})$ for all $s \in B_i$.

$$
\begin{aligned}
\mathbb{E}[f(s_i \mid S_{i-1}) \mid S_{i-1}] &= \mathbb{E}[f(s_i \mid S_{i-1}) \mid S_{i-1}, X_i] \Pr[X_i] \\
&\quad + \mathbb{E}[f(s_i \mid S_{i-1}) \mid S_{i-1}, X_i^c] \Pr[X_i^c] \\
&\geq \mathbb{E}[f(s_i \mid S_{i-1}) \mid S_{i-1}, X_i] \Pr[X_i] \\
&\geq \frac{\lambda_i}{k} \left( 1 - \frac{1}{e} \right) \cdot \frac{\sum_{\omega \in S_* \setminus S_{i-1}} f(\omega \mid S_{i-1})}{|S_* \setminus S_{i-1}|} \\
&= \frac{\lambda_i}{|S_* \setminus S_{i-1}|} \left( 1 - \frac{1}{e} \right) \cdot \frac{1}{k} \cdot \sum_{\omega \in S_* \setminus S_{i-1}} f(\omega \mid S_{i-1}) \\
&= \left( 1 - \frac{1}{e} \right) \cdot \frac{1}{k} \cdot \sum_{\omega \in S_* \setminus S_{i-1}} f(\omega \mid S_{i-1}) \\
&\geq \left( 1 - \frac{1}{e} \right) \cdot \frac{1}{k} \cdot \gamma \cdot (f(S_{i-1} \cup S_*) - f(S_{i-1})) \\
&\geq \left( 1 - \frac{1}{e} \right) \cdot \frac{1}{k} \cdot \gamma \cdot (\mathrm{OPT} - f(S_{i-1}))
\end{aligned}
$$

For the last 2 inequalities, we use the fact that $f(S) = \log \det(\boldsymbol{L}_S)$ is monotone non-decreasing and has a submodularity ratio of $\gamma = \left( 2 \frac{\log \sigma_{\max}}{\log \sigma_{\min}} - 1 \right)^{-1}$ when $\sigma_{\min} > 1$ (Gartrell et al., 2021)[Eq. 45].

Taking expectation over all random draws of $S_{i-1}$, we get

$$
\mathbb{E}[f(s_i \mid S_{i-1})] \geq \left( 1 - \frac{1}{e} \right) \cdot \frac{\gamma}{k} (\mathrm{OPT} - \mathbb{E}[f(S_{i-1})])
$$

Combining the above equation with $f(s_i \mid S_{i-1}) = f(S_i) - f(S_{i-1})$, we have

$$
\mathbb{E}[f(S_i)] - \mathbb{E}[f(S_{i-1})] \geq \left( 1 - \frac{1}{e} \right) \cdot \frac{\gamma}{k} \cdot (\mathrm{OPT} - \mathbb{E}[f(S_{i-1})])
$$

Next we have

$$
-(\mathrm{OPT} - \mathbb{E}[f(S_i)]) + (\mathrm{OPT} - \mathbb{E}[f(S_{i-1})]) \geq \left( 1 - \frac{1}{e} \right) \cdot \frac{\gamma}{k} \cdot (\mathrm{OPT} - \mathbb{E}[f(S_{i-1})])
$$

Re-organizing the above equation, we obtain

$$
\mathrm{OPT} - \mathbb{E}[f(S_i)] \leq \left( 1 - \left( 1 - \frac{1}{e} \right) \cdot \frac{\gamma}{k} \right) (\mathrm{OPT} - \mathbb{E}[f(S_{i-1})])
$$

Applying the above equation recursively $k$ times,

$$
\begin{aligned}
\mathrm{OPT} - \mathbb{E}[f(S_k)] &\leq \left( 1 - \left( 1 - \frac{1}{e} \right) \cdot \frac{\gamma}{k} \right)^k (\mathrm{OPT} - \mathbb{E}[f(S_0)]) \\
&= \left( 1 - \left( 1 - \frac{1}{e} \right) \cdot \frac{\gamma}{k} \right)^k \mathrm{OPT}
\end{aligned}
$$

where the last step follows from $f(S_0) = 0$.

Re-organized the terms again, we have

$$
\begin{aligned}
\mathbb{E}[f(S_k)] &\geq \left( 1 - \left( 1 - \left( 1 - \frac{1}{e} \right) \cdot \frac{\gamma}{k} \right)^k \right) \mathrm{OPT} \\
&\geq \left( 1 - e^{-\gamma(1 - \frac{1}{e})} \right) \mathrm{OPT}
\end{aligned}
$$

When we substitute $\gamma = \left(2\frac{\log \sigma_{\max}}{\log \sigma_{\min}} - 1\right)^{-1}$, we get our final inequality:

$$\mathbb{E}[f(S_k)] \geq \left(1 - \frac{1}{\sigma_{\min}^{(1-\frac{1}{e})\cdot(2\log \sigma_{\max} - \log \sigma_{\min})}}\right) \text{OPT}$$

$\blacksquare$

**Theorem 3.** *For any length-$n$ stream $(\boldsymbol{v}_1, \boldsymbol{b}_1), \ldots, (\boldsymbol{v}_n, \boldsymbol{b}_n)$ where $(\boldsymbol{v}_t, \boldsymbol{b}_t) \in \mathbb{R}^d \times \mathbb{R}^d \, \forall \, t \in [n]$, the worst-case update time of Algorithm 1 is $O(\mathcal{T}_{\det}(k,d))$ where $\mathcal{T}_{\det}(k,d)$ is the time taken to compute $f(S) = \det(\boldsymbol{V}_S^\top \boldsymbol{V} + \boldsymbol{B}_S^\top \boldsymbol{C}\boldsymbol{B})$ for $|S| = k$. The total time taken is $O(n \cdot \mathcal{T}_{\det}(k,d))$ and the space used at any time step is $O(k^2 + d^2)$.*

*Proof.* For any particular data-point $(\boldsymbol{v}_t, \boldsymbol{b}_t)$, Algorithm 1 needs to compute $f(S_{i-1} \cup \{t\})$, which takes at most $\mathcal{T}_{\det}(k,d)$ time. All the other comparison and update steps are much faster and so the worst-case update time is $O(\mathcal{T}_{\det}(k,d))$. The space needed to compute $\det(\boldsymbol{V}_S^\top \boldsymbol{V} + \boldsymbol{B}_S^\top \boldsymbol{C}\boldsymbol{B}_S)$ is at most $O(k^2 + d^2)$ where $S = S_{i-1} \cup \{t\}$. The algorithm also needs to store $S_{i-1}, s_i$ and $f(S_{i-1} \cup \{s_i\})$ but all of these are dominated by $O(k^2 + d^2)$ space needed to compute the determinant. $\blacksquare$

## B  ONLINE MAP INFERENCE DETAILS

**Theorem 5.** *For any length-$n$ stream $(\boldsymbol{v}_1, \boldsymbol{b}_1), \ldots, (\boldsymbol{v}_n, \boldsymbol{b}_n)$ where $(\boldsymbol{v}_t, \boldsymbol{b}_t) \in \mathbb{R}^d \times \mathbb{R}^d \, \forall \, t \in [n]$, the worst case update time of Algorithm 2 is $O(\mathcal{T}_{\det}(k,d) \cdot k \log^2(\Delta))$ where $\mathcal{T}_{\det}(k,d)$ is the time taken to compute $f(S) = \det(\boldsymbol{V}_S^\top \boldsymbol{V} + \boldsymbol{B}_S^\top \boldsymbol{C}\boldsymbol{B})$ for $|S| = k$. Furthermore, the amortized update time is $O(\mathcal{T}_{\det}(k,d) \cdot (k + \frac{k \log^2(\Delta)}{n}))$ and the space used at any time step is at most $O(k^2 + d^2 + d \log_\alpha(\Delta))$.*

*Proof.* For every iteration of the while loop in line 4: It takes at most $\mathcal{T}_{\det}(k,d)$ time for checking the first if condition (lines 5-6). The $\arg\max_{j \in S} f(S \cup \{t\} \setminus \{j\})$ step takes at most $k \cdot \mathcal{T}_{\det}(k,d)$ time. The while loop in line 12 takes time at most $|S| \cdot |T| \cdot \mathcal{T}_{\det}(k,d)$ for every instance of an increase in $f(S)$. Note that $f(S)$ can increase at most $\log_\alpha(\Delta)$ times since the value of $f(S)$ cannot exceed $\text{OPT}_k$. Therefore, the update time of Algorithm 2 is at most $\mathcal{T}_{\det}(k,d) + k \cdot \mathcal{T}_{\det}(k,d) + \log_\alpha(\Delta) \cdot (|S| \cdot |T| \cdot \mathcal{T}_{\det}(k,d)) \leq \mathcal{T}_{\det}(k,d) \cdot (k + 1 + k \log_\alpha^2(\Delta))$ since $|S| \leq k$ and $|T| \leq \log_\alpha(\Delta)$. Notice that the cardinality of $T$ can increase by 1 only when the value of $f(S)$ increases at least by a factor of $\alpha$ and so $|T| \leq \log_\alpha(\Delta)$.

During any time step $t$, the algorithm needs to store the indices in $S, T$ and the corresponding matrices $\boldsymbol{V}_S, \boldsymbol{B}_S, \boldsymbol{V}_T, \boldsymbol{B}_T$. Since $|S| \leq k, |T| \leq \log_\alpha(\Delta)$ and it takes $d$ words to store every $\boldsymbol{v}_i$ and $\boldsymbol{b}_i$, we need at most $k + \log_\alpha(\Delta) + 2dk + 2d \log_\alpha(\Delta)$ words to store all these in memory. The space needed to compute $\det(\boldsymbol{V}_S^\top \boldsymbol{V}_S + \boldsymbol{B}_S^\top \boldsymbol{C}\boldsymbol{B}_S)$ is at most $O(k^2 + d^2)$. We compute all such determinants one after the other in our algorithm. So the algorithm only needs space for one such computation during it's run. Therefore, the space required by Algorithm 2 is $O(k^2 + d^2 + d \log_\alpha(\Delta))$. $\blacksquare$

**Theorem 7.** *For any length-$n$ stream $(\boldsymbol{v}_1, \boldsymbol{b}_1), \ldots, (\boldsymbol{v}_n, \boldsymbol{b}_n)$ where $(\boldsymbol{v}_t, \boldsymbol{b}_t) \in \mathbb{R}^d \times \mathbb{R}^d \, \forall \, t \in [n]$, the worst case update time of Algorithm 3 is $O(\mathcal{T}_{\det}(k,d) \cdot k^2 \log^3(\Delta))$ where $\mathcal{T}_{\det}(k,d)$ is the time taken to compute $f(S) = \det(\boldsymbol{V}_S^\top \boldsymbol{V} + \boldsymbol{B}_S^\top \boldsymbol{C}\boldsymbol{B})$ for $|S| = k$. The amortized update time is $O\left(\mathcal{T}_{\det}(k,d) \cdot \left(k^2 + \frac{k^2 \log^3(\Delta)}{n}\right)\right)$ and the space used at any time step is at most $O(k^2 + d^2 + d \log_\alpha(\Delta))$.*

*Proof.* It takes at most $\mathcal{T}_{\det}(k,d)$ time for lines 5-6 (same as in LSS). The $\arg\max_{a,b \in S} \left(f(S \cup \{t\} \setminus \{a\}), f(S \cup \{t-1, t\} \setminus \{a, b\})\right)$ step takes at most $k^2 \cdot \mathcal{T}_{\det}(k,d)$ time. The while loop in line 18 takes time at most $|S|^2 \cdot |T|^2 \cdot \mathcal{T}_{\det}(k,d)$ for every instance of an increase in $f(S)$. Similar to LSS, $f(S)$ can increase at most by a factor of $\log_\alpha(\Delta)$ since the value of $f(S)$ cannot exceed $\text{OPT}_k$. Therefore, the update time of Algorithm 3 is at most $\mathcal{T}_{\det}(k,d) + k^2 \cdot \mathcal{T}_{\det}(k,d) + \log_\alpha(\Delta) \cdot \left(|S|^2 \cdot |T|^2 \cdot \mathcal{T}_{\det}(k,d)\right) \leq \mathcal{T}_{\det}(k,d) \cdot \left(k^2 + 1 + k^2 \log_\alpha^3(\Delta)\right)$ since $|S| \leq k$ and $|T| \leq \log_\alpha(\Delta)$.

---

**Algorithm 5** ONLINE-GREEDY: Online Greedy MAP Inference for NDPPs

1: **Input:** A stream of data points $\{(\boldsymbol{v}_1, \boldsymbol{b}_1), (\boldsymbol{v}_2, \boldsymbol{b}_2), \ldots, (\boldsymbol{v}_n, \boldsymbol{b}_n)\}$
2: **Output:** A solution set $S$ of cardinality $k$ at the end of the stream.
3: $S \leftarrow \emptyset$
4: **while** new data $(\boldsymbol{v}_t, \boldsymbol{b}_t)$ arrives in stream at time $t$ **do**
5:     **if** $|S| < k$ and $f(S \cup \{t\}) \neq 0$ **then**
6:         $S \leftarrow S \cup \{t\}$
7:     **else**
8:         $i \leftarrow \arg\max_{j \in S} f(S \cup \{t\} \setminus \{j\})$
9:         **if** $f(S \cup \{t\} \setminus \{i\}) > f(S)$ **then**
10:             $S \leftarrow S \cup \{t\} \setminus \{i\}$
11: **return** $S$

---

Although Algorithm 3 executes more number of determinant computations than Algorithm 2, all of them are done sequentially and only the maximum value among all the previously computed values in any specific iteration needs to be stored in memory. Therefore, the space needed is (nearly) the same for both the algorithms. ∎

## C    HARD INSTANCE FOR ONLINE MAP INFERENCE OF NDPPS

**Outline**: We will now give a high-level description of a hard instance for online MAP inference of NDPPs (this is inspired by (Anari & Vuong, 2021, Example 5)) . Suppose we have a total of $2d$ items consisting of **pairs** of complementary items like modem-router, printer-ink cartridge, pencil-eraser etc. Let us use $\{1, 1^\mathsf{c}, 2, 2^\mathsf{c}, \ldots, d, d^\mathsf{c}\}$ to denote them. Any item $i$ is independent of every item other than it's complement $i^\mathsf{c}$. Individually, $\Pr[\{i\}] = \Pr[\{i^\mathsf{c}\}] = 0$ . And $\Pr[\{i, i^\mathsf{c}\}] = x_i^2$ with $x_i > 0$ for all $i \in [d]$. Also, we have $\Pr[\{i, j\}] = 0$ for any $i \neq j$. Suppose any of our online algorithms are given the sequence $\{1, 2, 3, \ldots, d, r^\mathsf{c}\}$ where $r \in [d]$ is some arbitrary item and the algorithm needs to pick 2 items i.e. $k = 2$. Then, OPT $> 0$ whereas all of our online algorithms (Online LSS, Online 2-neighbor, Online-Greedy) will fail to output a valid solution.

**Details**: Let $0 < x_1 < x_2 < \cdots < x_d$. Suppose

$$
\boldsymbol{C} = \begin{bmatrix}
0 & x_1 & & & & & \\
-x_1 & 0 & & & & & \\
& & 0 & x_2 & & & \\
& & -x_2 & 0 & & & \\
& & & & \ddots & & \\
& & & & & 0 & x_d \\
& & & & & -x_d & 0
\end{bmatrix}
$$

$\boldsymbol{C} \in \mathbb{R}^{2d \times 2d}$ is a skew-symmetric (i.e. $\boldsymbol{C} = -\boldsymbol{C}^\top$) block diagonal matrix where the blocks are of the form $\begin{bmatrix} 0 & x_i \\ -x_i & 0 \end{bmatrix}$. Suppose we have a total of $2d$ items consisting of $d$ pairs of complementary items. We use $\{1, 1^\mathsf{c}, 2, 2^\mathsf{c}, \ldots, d, d^\mathsf{c}\}$ to denote them. Let $\boldsymbol{v}_i = \boldsymbol{v}_{i^\mathsf{c}} = \boldsymbol{0} \ \forall \ i \in [d]$ and $\boldsymbol{b}_1 = \boldsymbol{e}_1, \boldsymbol{b}_{1^\mathsf{c}} = \boldsymbol{e}_2, \ldots, \boldsymbol{b}_{d^\mathsf{c}} = \boldsymbol{e}_{2d}$ where $\boldsymbol{e}_1, \boldsymbol{e}_2, \ldots, \boldsymbol{e}_{2d}$ are the standard unit vectors in $\mathbb{R}^{2d}$ i.e. $B = \boldsymbol{I}_{2d}$.

For a pair of complementary items $S = \{i, i^\mathsf{c}\}, f(S) = x_i^2$. Without loss of generality, consider $S = \{1, 1^\mathsf{c}\}$. Then we can compute $\boldsymbol{B}_S^\top \boldsymbol{C} \boldsymbol{B}_S$ as follows:

$$
\begin{aligned}
\boldsymbol{B}_S^\top \boldsymbol{C} \boldsymbol{B}_S &= \begin{bmatrix} e_1 & e_2 \end{bmatrix}^\top \boldsymbol{C} \begin{bmatrix} e_1 & e_2 \end{bmatrix} \\
&= \begin{bmatrix} 0 & x_1 & 0 & 0 & \cdots & 0 \\ -x_1 & 0 & 0 & 0 & \cdots & 0 \end{bmatrix} \cdot \begin{bmatrix} e_1 & e_2 \end{bmatrix} \\
&= \begin{bmatrix} 0 & x_1 \\ -x_1 & 0 \end{bmatrix}
\end{aligned}
$$

In this case, we have $f(S) = x_1^2$.

For any pair of non-complementary items $S = \{i_1, i_2\}$ where the indices are distinct, $f(S) = 0$. Without loss of generality, we can consider $S = \{1, 2\}$. Then,

$$
\begin{aligned}
\boldsymbol{B}_S^\top \boldsymbol{C} \boldsymbol{B}_S &= [e_1 \quad e_3]^\top \boldsymbol{C} [e_1 \quad e_3] \\
&= \begin{bmatrix} 0 & x_1 & 0 & 0 & \cdots & 0 \\ 0 & 0 & 0 & x_2 & \cdots & 0 \end{bmatrix} \cdot [e_1 \quad e_3] \\
&= \begin{bmatrix} 0 & 0 \\ 0 & 0 \end{bmatrix}
\end{aligned}
$$

And so, we have that $f(S) = 0$.

## D  LEARNING DETAILS

Our approximate objective

$$
\psi_t(\boldsymbol{V}, \boldsymbol{B}, \boldsymbol{C}) = \log\det\left(\boldsymbol{V}_{S_t}^\top \boldsymbol{V}_{S_t} + \boldsymbol{B}_{S_t}^\top \boldsymbol{C} \boldsymbol{B}_{S_t}\right) - Z(\boldsymbol{V}_{S_t}, \boldsymbol{B}_{S_t}, \boldsymbol{C}) - R(\boldsymbol{V}_{S_t}, \boldsymbol{B}_{S_t}) \tag{5}
$$

We will now provide a derivation of i.e. $\nabla_{\boldsymbol{V}_{S_t}, \boldsymbol{B}_{S_t}, \boldsymbol{C}}\, \psi_t$. Let us first look at the gradient of the first term

$$
\nabla_{\boldsymbol{V}_{S_t}} \log\det\left(\boldsymbol{V}_{S_t}^\top \boldsymbol{V}_{S_t} + \boldsymbol{B}_{S_t}^\top \boldsymbol{C} \boldsymbol{B}_{S_t}\right) = 2\boldsymbol{V}_{S_t}\left(\boldsymbol{V}_{S_t}^\top \boldsymbol{V}_{S_t} + \boldsymbol{B}_{S_t}^\top \boldsymbol{C} \boldsymbol{B}_{S_t}\right)^{-1}
$$

$$
\nabla_{\boldsymbol{B}_{S_t}} \log\det\left(\boldsymbol{V}_{S_t}^\top \boldsymbol{V}_{S_t} + \boldsymbol{B}_{S_t}^\top \boldsymbol{C} \boldsymbol{B}_{S_t}\right) = 2\boldsymbol{C}\boldsymbol{B}_{S_t}\left(\boldsymbol{V}_{S_t}^\top \boldsymbol{V}_{S_t} + \boldsymbol{B}_{S_t}^\top \boldsymbol{C} \boldsymbol{B}_{S_t}\right)^{-1}
$$

$$
\nabla_{\boldsymbol{C}} \log\det\left(\boldsymbol{V}_{S_t}^\top \boldsymbol{V}_{S_t} + \boldsymbol{B}_{S_t}^\top \boldsymbol{C} \boldsymbol{B}_{S_t}\right) = \boldsymbol{B}_{S_t}\left(\boldsymbol{V}_{S_t}^\top \boldsymbol{V}_{S_t} + \boldsymbol{B}_{S_t}^\top \boldsymbol{C} \boldsymbol{B}_{S_t}\right)^{-1} \boldsymbol{B}_{S_t}^\top
$$

Now, we consider the gradient of $Z(\boldsymbol{V}_{S_t}, \boldsymbol{B}_{S_t}, \boldsymbol{C})$, which consists of three parts: $\nabla_{\boldsymbol{V}_{S_t}} Z, \nabla_{\boldsymbol{B}_{S_t}} Z, \nabla_{\boldsymbol{C}} Z$. Let $\boldsymbol{X} = \boldsymbol{I} - \boldsymbol{V}_{S_t}^\top(\boldsymbol{I}_d + \boldsymbol{V}_{S_t}\boldsymbol{V}_{S_t}^\top)^{-1}\boldsymbol{V}_{S_t}$. Note that

$$
\boldsymbol{Z} = \log\det\left(\boldsymbol{I}_d + \boldsymbol{V}_{S_t}\boldsymbol{V}_{S_t}^\top\right) + \log\det\left(\boldsymbol{C}^{-1} + \boldsymbol{B}_{S_t}(\boldsymbol{I} - \boldsymbol{V}_{S_t}^\top(\boldsymbol{I}_d + \boldsymbol{V}_{S_t}\boldsymbol{V}_{S_t}^\top)^{-1}\boldsymbol{V}_{S_t})\boldsymbol{B}_{S_t}^\top\right) + \log\det(\boldsymbol{C})
$$

We refer the reader to (Gartrell et al., 2021, Eqn 19) for a derivation of the above form of $\boldsymbol{Z}$. Then,

$$
\begin{aligned}
\nabla_{\boldsymbol{V}_{S_t}} Z &= 2\boldsymbol{V}_{S_t}^\top(\boldsymbol{I}_d + \boldsymbol{V}_{S_t}\boldsymbol{V}_{S_t}^\top)^{-1} \\
&\quad - \boldsymbol{X}\boldsymbol{B}_{S_t}^\top((\boldsymbol{C}^{-1} + \boldsymbol{B}_{S_t}\boldsymbol{X}\boldsymbol{B}_{S_t}^\top)^{-1} + (\boldsymbol{C}^{-\top} + \boldsymbol{B}_{S_t}\boldsymbol{X}\boldsymbol{B}_{S_t}^\top)^{-1})\boldsymbol{B}_{S_t}\boldsymbol{X}\boldsymbol{V}_{S_t}^\top
\end{aligned} \tag{6}
$$

$$
\nabla_{\boldsymbol{B}_{S_t}} Z = \boldsymbol{X}\boldsymbol{B}_{S_t}^\top\left((\boldsymbol{C}^{-1} + \boldsymbol{B}_{S_t}\boldsymbol{X}\boldsymbol{B}_{S_t}^\top)^{-1} + (\boldsymbol{C}^{-\top} + \boldsymbol{B}_{S_t}\boldsymbol{X}\boldsymbol{B}_{S_t}^\top)^{-1}\right) \tag{7}
$$

$$
\nabla_{\boldsymbol{C}} Z = \boldsymbol{C}^{-\top} - \boldsymbol{C}^{-\top}(\boldsymbol{C}^{-1} + \boldsymbol{B}_{S_t}\boldsymbol{X}\boldsymbol{B}_{S_t}^\top)^{-\top}\boldsymbol{C}^{-\top} \tag{8}
$$

Since we only have one subset at a time step, $R(\boldsymbol{V}_{S_t}, \boldsymbol{B}_{S_t}) = \alpha \sum_{i \in S_t} \|\boldsymbol{v}_i\|^2 + \beta \sum_{i \in S_t} \|\boldsymbol{b}_i\|^2$. Therefore, $\nabla_{\boldsymbol{V}_{S_t}} R = 2\alpha\boldsymbol{V}_{S_t}, \nabla_{\boldsymbol{B}_{S_t}} R = 2\beta\boldsymbol{B}_{S_t}$.

We can then substitute $\nabla\boldsymbol{Z}$ from Equations 6,7,8 in the following equations to get the gradient required for updates $\nabla\psi_t$:

$$
\nabla_{\boldsymbol{V}_{S_t}} \psi_t = 2\boldsymbol{V}_{S_t}\left(\boldsymbol{V}_{S_t}^\top \boldsymbol{V}_{S_t} + \boldsymbol{B}_{S_t}^\top \boldsymbol{C} \boldsymbol{B}_{S_t}\right)^{-1} - \nabla_{\boldsymbol{V}_{S_t}} Z - 2\alpha\boldsymbol{V}_{S_t} \tag{9}
$$

$$
\nabla_{\boldsymbol{B}_{S_t}} \psi_t = 2\boldsymbol{C}\boldsymbol{B}_{S_t}\left(\boldsymbol{V}_{S_t}^\top \boldsymbol{V}_{S_t} + \boldsymbol{B}_{S_t}^\top \boldsymbol{C} \boldsymbol{B}_{S_t}\right)^{-1} - \nabla_{\boldsymbol{B}_{S_t}} Z - 2\beta\boldsymbol{B}_{S_t} \tag{10}
$$

$$
\nabla_{\boldsymbol{C}} \psi_t = \boldsymbol{B}_{S_t}\left(\boldsymbol{V}_{S_t}^\top \boldsymbol{V}_{S_t} + \boldsymbol{B}_{S_t}^\top \boldsymbol{C} \boldsymbol{B}_{S_t}\right)^{-1} \boldsymbol{B}_{S_t}^\top - \nabla_{\boldsymbol{C}} Z \tag{11}
$$

**Theorem 9.** *Algorithm 4 uses $O(d'^2 + d^2)$ space $^\ddagger$. Also, the update time for a single subset $S_t$ arriving at time $t$ in the stream is $O(d^3 + d'^3)$ where $d'$ is the size of the largest set in the stream.*

---

$^\ddagger$Note that the $O(nd + d^2)$ required to store the matrices $\boldsymbol{V}$, $\boldsymbol{B}$, and $\boldsymbol{C}$ are omitted for simplicity (since these terms are common to all algorithms).

*Proof.* To compute $\nabla_{\boldsymbol{V}_{S_t}} Z$, it takes space at most $O(d'^2 + d^2)$ as it involves matrix multiplications and inversions of matrices of sizes $d' \times d, d \times d, d' \times d'$. Similarly for $\nabla_{\boldsymbol{B}_{S_t}} Z, \nabla_{\boldsymbol{C}} Z$. Computing the gradient of the first term in Equation 4 requires inversion of a $d' \times d'$ matrix, which takes at most $O(d'^2)$ space. And the gradient of the sum of norms term in Equation 4 takes at most $O(d'd)$ space to compute.

As described in the proof of the previous lemma, we will need to compute matrix multiplications and inversions of size at most $d \times d, d' \times d'$, and $d' \times d$. All of these can be done in time $O(d^3 + d'^3)$. ∎

**Theorem 10.** *Given a stream of subsets $\{S_1, S_2, \ldots, S_m\}$ of length-$m$, Algorithm 4 learns an NDPP kernel $\boldsymbol{L} = \boldsymbol{V}^\top \boldsymbol{V} + \boldsymbol{B}^\top \boldsymbol{C} \boldsymbol{B}$ of rank-$d$ using only a single-pass over the stream in $O(md'^3 + md^3)$ time where $d' = \max_t(|S_t|) \ll n$. Note $\boldsymbol{V}, \boldsymbol{B} \in \mathbb{R}^{d \times n}, \boldsymbol{C} \in \mathbb{R}^{d \times d}$, and $d \ll n$.*

In comparison, the offline learning algorithm (Gartrell et al., 2021) takes at most $T_{\max}$ passes over the entire data ($m$ subsets), and since all $m$ subsets are stored in memory (offline, non-streaming), they use all available data at once to update the embedding matrices of the NDPP model. Notice that $\log \det\left(\boldsymbol{V}_{S_t}^\top \boldsymbol{V}_{S_t} + \boldsymbol{B}_{S_t}^\top \boldsymbol{C} \boldsymbol{B}_{S_t}\right)$ in Eq. 3 is computed for every subset, thus taking $O(md')$ time for all $m$ sets in the stream where $d'$ is the size of the largest set in the stream. Furthermore, $\log \det\left(\boldsymbol{V}^\top \boldsymbol{V} + \boldsymbol{B}^\top \boldsymbol{C} \boldsymbol{B} + \boldsymbol{I}\right)$ (2nd term in Eq. 3) takes $O(nd^2)$ time where $n \ll m$, and this can be computed a constant number of times with respect to $m$. Similarly for the regularization term $R(\boldsymbol{V}, \boldsymbol{B})$.

## E  DATASETS

We use several real-world datasets composed of sets (or baskets) of items from some ground set of items.

- **UK Retail:** This is an online retail dataset consisting of sets of items all purchased together by users (in a single transaction) (Chen et al., 2012). There are 19,762 transactions (sets of items purchased together) that consist of 3,941 items. Transactions with more than 100 items are discarded.

- **MovieLens:** This dataset contains sets of movies that users watched (Sharma et al., 2019). There are 29,516 sets consisting of 12,549 movies.

- **Amazon Apparel**: This dataset consists of 14,970 registries (sets) from the apparel category of the Amazon Baby Registries dataset, which is a public dataset that has been used in prior work on NDPPs (Gartrell et al., 2021; 2019). These apparel registries are drawn from 100 items in the apparel category.

- **Amazon 3-category**: We also use a dataset composed of the apparel, diaper, and feeding categories from Amazon Baby Registries, which are the most popular categories, giving us 31,218 registries made up of 300 items (Gartrell et al., 2019).

- **Instacart:** This dataset represents sets of items purchased by users on Instacart (Instacart, 2017). Sets with more than 100 items are ignored. This gives 3.2 million total item-sets from 49,677 unique items.

- **Million Song:** This is a dataset of song playlists put together by users where every playlist is a set (basket) of songs played by Echo Nest users (McFee et al., 2012). Playlists with more than 150 songs are discarded. This gives 968,674 playlists from 371,410 songs.

- **Customer Dashboards:** This dataset consists of dashboards or baskets of visualizations created by users (Qian et al., 2020). Each dashboard represents a set of visualizations selected by a user. There are 63436 dashboards (baskets/sets) consisting of 1206 visualizations.

- **Web Logs:** This dataset consists of sets of webpages (baskets) that were all visited in the same session. There are 2.8 million baskets (sets of webpages) drawn from 2 million webpages.

- **Company Retail:** This dataset contains the set of items viewed (or purchased) by a user in a given session. Sets (baskets) with more than 100 items are discarded. This results in 2.5 million baskets consisting of 107,349 items.

The last two datasets are proprietary company data.

# F    ADDITIONAL EXPERIMENTAL RESULTS

## F.1    NUMBER OF DETERMINANT COMPUTATIONS

We perform several experiments comparing the number of determinant computations (as a system-independent proxy for time) of all our online algorithms. We do not compare with offline greedy here because that algorithm doesn't explicitly compute all the determinants. Results comparing the number of determinant computations as a function of the number of data points analyzed for a variety of datasets are provided in Figure 3. Online-2-neighbor requires the most number of determinant computations but also gives the best results in terms of solution value. Online-LSS and Online-Greedy use very similar number of determinant computations.

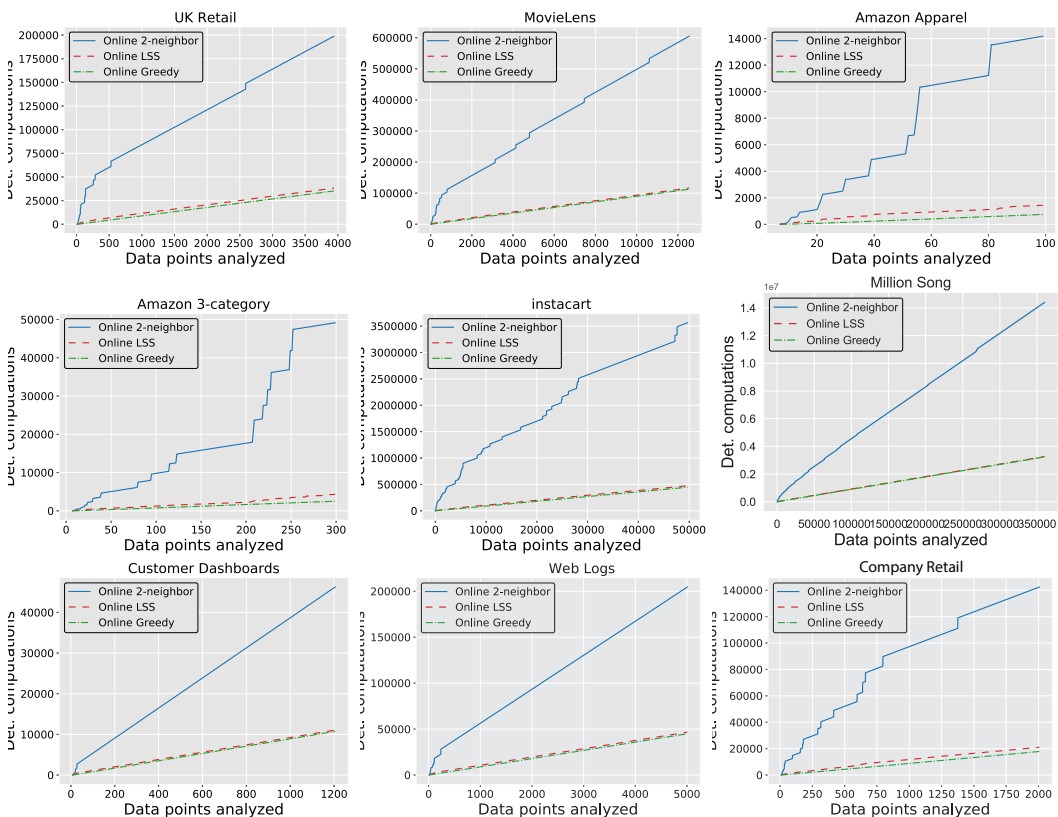

Figure 3:  Results comparing the number of determinant computations as a function of the number of data points analyzed for all our online algorithms. Online-2-neighbor requires the most number of determinant computations but also gives the best results in terms of solution value. Online-LSS and Online-Greedy use very similar number of determinant computations.

## F.2 Number of Swaps

Results comparing the number of swaps (as a measure of solution consistency) of all our online algorithms can be found in Figure 4. Online-Greedy has the most number of swaps and therefore the least consistent solution set. On most datasets, the number of swaps by Online-2-neighbor is very similar to Online-LSS.

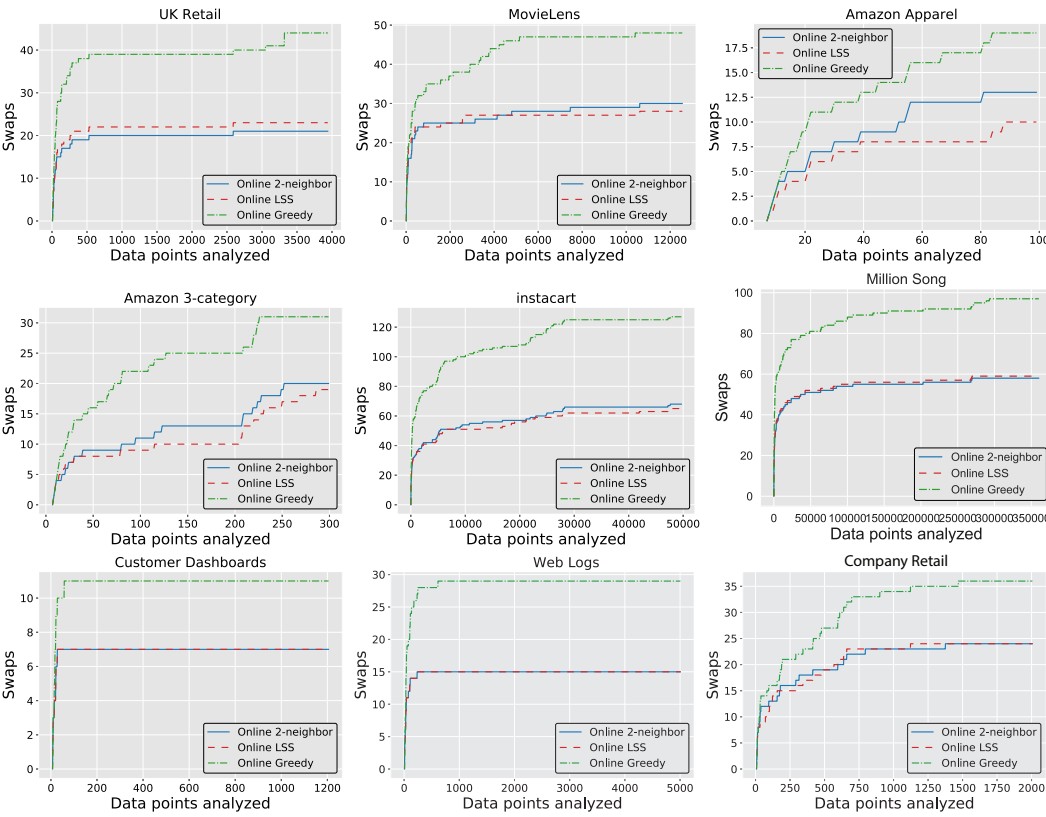

Figure 4: Results comparing the number of swaps of all our online algorithms. Online-Greedy does the most number of swaps and therefore has the least consistent solution set. On most datasets, the number of swaps by Online-2-neighbor is very similar to Online-LSS.

## F.3 Random Streams

We also investigate our algorithms under the random stream paradigm. For this setting, we use some of the previous real-world datasets, and randomly permute the order in which the data appears in the stream. We do this 100 times and report the average of solution values in Figure 5 and the average of number of determinant computations and swaps in Figure 6. We observe that Online-2-neighbor and Online-LSS give very similar performance in this regime and they are always better than Online-Greedy.

## F.4 Ablation study varying $\epsilon$

To study the effect of $\epsilon$ in Online-LSS (Algorithm 2), we vary $\epsilon \in \{0.05, 0.1, 0.3, 0.5\}$ and analyze the value of the obtained solutions, number of determinant computations, and number of swaps. We notice that, in general, the solution quality, number of determinant computations, and the number of swaps increase as $\epsilon$ decreases. Results are provided in Figure 7.

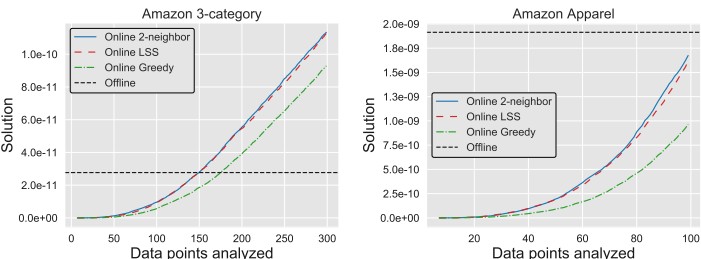

Figure 5: Solution quality as a function of the number of data points analyzed in the random stream paradigm. Online-2-neighbor and Online-LSS give very similar performance in this setting and they are always better than Online-Greedy.

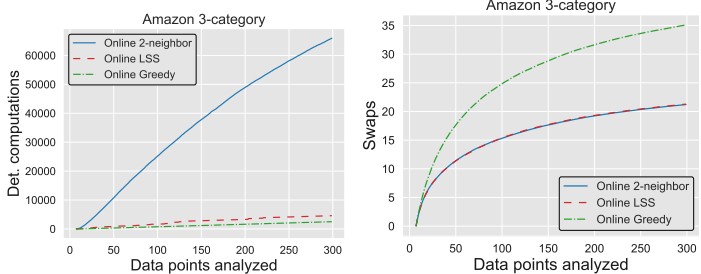

Figure 6: Number of determinant computations and swaps as a function of the number of data points analyzed in the random stream setting. Online-2-Neighbor needs more determinant computations than Online-LSS but has very similar number of swaps in this setting. Note that $\epsilon = \alpha - 1$
.

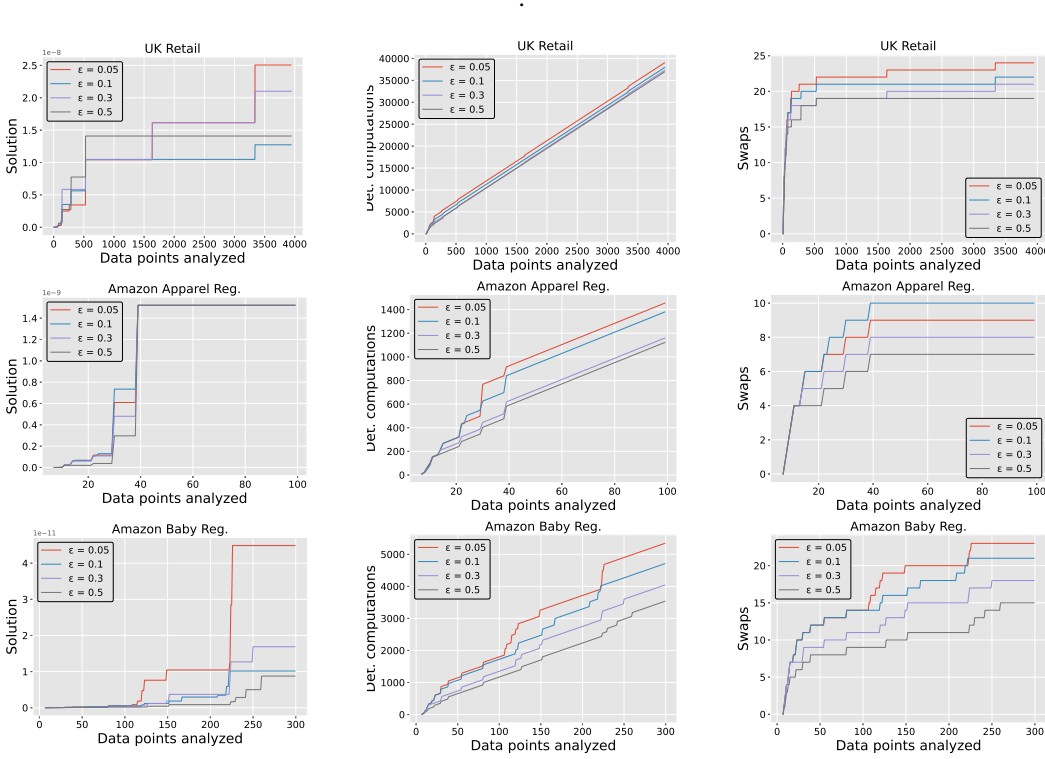

Figure 7: Performance of Online-LSS varying $\epsilon$ for $k = 8$. Solution quality, number of determinant computations, and number of swaps seem to increase with decreasing $\epsilon$.

### F.5 ABLATION STUDY VARYING SET SIZE $k$ AND $\epsilon$

In this set of experiments on the Amazon-Apparel dataset using Online-LSS, we study the choice of set size $k$ and $\epsilon$ on the solution quality, number of determinant computations, and number of swaps while fixing all other settings to be same as those used previously in Figure 5. We can see that as $k$ increases, the solution value decreases across all values of $\epsilon$. This is in accordance with our intuition that the probability of larger sets should be smaller. In general, the number of determinant computations and swaps increases for increasing $k$ across different values of $\epsilon$.

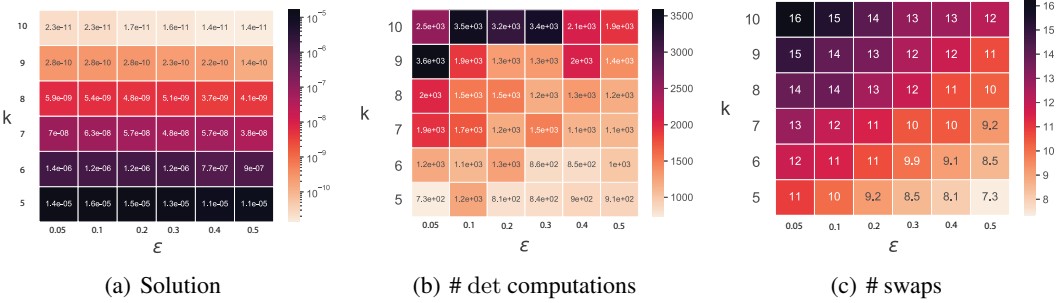

(a) Solution             (b) # det computations             (c) # swaps

Figure 8: Comparing the effect of set size $k$ and $\epsilon$ on the solution, number of determinant computations, and number of swaps for ONLINE-LSS.

### F.6 ONLINE LEARNING RESULTS

Figure 9 shows that the heat maps of the kernel matrices learned by our online learning algorithm and the offline learning algorithm are quite similar. We use the Amazon Apparel dataset for this set of experiments.

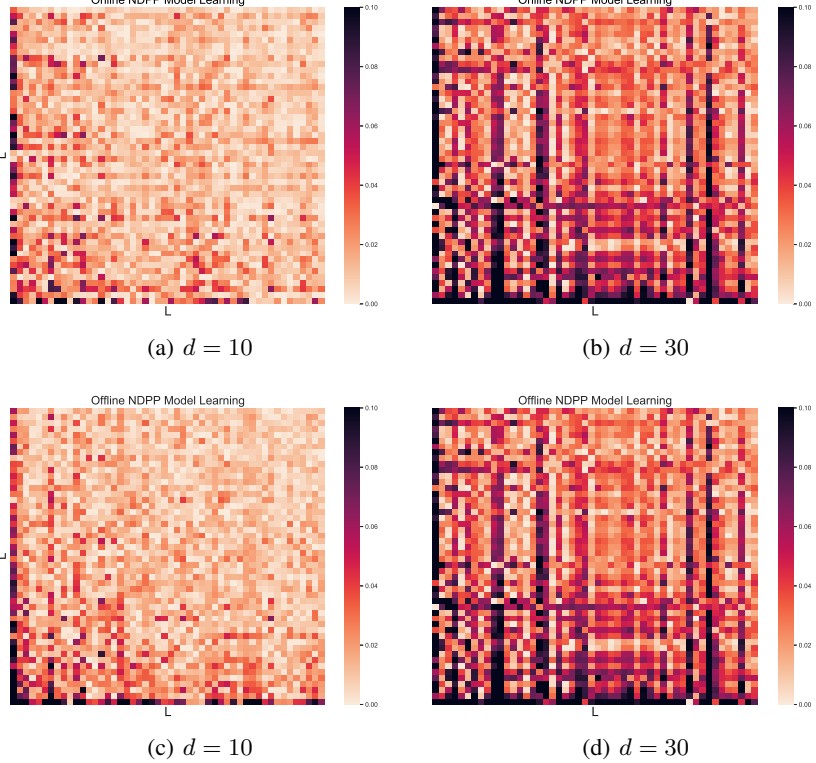

(a) $d = 10$             (b) $d = 30$

(c) $d = 10$             (d) $d = 30$

Figure 9: Heatmaps of the kernel matrices learned by our online algorithm (top) looks very similar to the ones learned offline using all data (bottom).

