# OpenReview forum: "Online MAP Inference and Learning for Nonsymmetric Determinantal Point Processes"
_ICLR.cc/2022/Conference — ICLR 2022 Submitted_

### Official Review · Reviewer_9kPN · 2021-11-02

**Correctness:** 2
**Technical Novelty And Significance:** 3
**Empirical Novelty And Significance:** 3
**Recommendation:** 5
**Confidence:** 4

**Main Review:**

Strengths:
- The proposed online and streaming algorithms for MAP inference and learning appear to be novel, and in a number of cases empirically outperform state-of-the-art offline NDPP algorithms for these tasks.
- The proposed streaming MAP inference algorithm (Alg. 1) has theoretical guarantees for MAP approximation quality, and time and space complexity.  The proposed online MAP inference algorithms (Algs. 2 and 4) have theoretical guarantees for time and space complexity.
- The proposed online learning algorithm (Alg. 3) has theoretical guarantees for time and space complexity.
- The paper is reasonably well written and easy to follow.

Weaknesses:
- Some of the claims made in Sec. 6.1 regarding the state-of-the-art NDPP learning algorithm in Gartrell et al. (2021) appear to be incorrect.  Sec. 6.1 claims that the Gartrell et al. (2021) learning algorithm: 1) must store all training data in memory; 2) must make multiple passes over the data; and 3) subsets are not processed sequentially as they arrive.  However, the implementation of the learning algorithm in Gartrell et al. (2021) uses Adam (a variant of SGD), which can run with a batch size of 1, or with minibatches, and thus can be run in a streaming setting that does not require all data to be loaded into memory.  Thus, claims 1 - 3 seem to be incorrect.
- The normalization term, $Z(V_{S_t}, B_{S_t}, C)$, used in the approximate objective (Eq. 4) for the online learning algorithm is not equivalent to the standard normalization term for a NDPP ($Z(V, B, C)$), nor is it clear that this “approximate” normalization term actually provides a reasonable approximation to the true NDPP normalizer.  The authors do not address this issue in the paper, and thus the proposed online learning algorithm has no theoretical approximation guarantees when compared to standard NDPP learning algorithms.  Therefore, optimizing Eq. 4 appears to violate the requirements described in Definition 8, since the true NDPP regularized log-likelihood is not being maximized.  This seems to be a critical issue.


**Summary Of The Paper:**

This paper proposes online and streaming algorithms for MAP inference and learning for nonsymmetric determinantal point processes (NDPPs).  For the streaming setting, data points arrive in an arbitrary order, and the algorithms are constrained to using a single pass over the data, along with requiring sublinear memory consumption.  In the online setting, there is the additional requirement of maintaining a valid solution at any time step.  The authors provide some theoretical guarantees for the proposed algorithms, and perform experiments that demonstrate that their performance is comparable to (or better than) offline algorithms for these tasks.


**Summary Of The Review:**

This paper has strong contributions in the area of streaming and online MAP inference algorithms.  However, there are some notable issues with the contributions regarding the online learning algorithm, including the comparison with prior work, and the correctness of the approximate optimization objective (Eq. 4), as described above.  Thus, unless the authors can address these issues in the rebuttal, it is hard to recommend this paper for acceptance.

---

> ### Author Response · Authors · 2021-11-23
> **Response to Reviewer 9kPN**
>
> Thank you for your review. We agree that if a batch size of 1 is used in the algorithm by Gartrell et.al (without any shuffling of the data for SGD) and also using only a single iteration, then the 3 claims about the learning algorithm of Gartrell et.al are incorrect. But that is not what they do in the paper (they use a batch size of 800 and take 1000 iterations over the entire data for Instacart with which we compare). Even if a batch size of 1 is used in the algorithm by Gartrell et.al, our algorithm will run much faster as it doesn’t need to update the representations of all the items in every time step (unlike the algorithm by Gartrell et.al). Please see the common response for more details.

---

> > ### Comment · Reviewer_9kPN · 2021-11-25
> > **Post-rebuttal feedback**
> >
> > I thank the authors for their rebuttal comments and the revision of their paper.  The rebuttal comments and changes have helped to clarify the contributions of this paper.  The proposed online and streaming MAP inference algorithms, and the online learning algorithm, clearly have strong empirical performance.  However, my concerns about the lack of theoretical guarantees and understanding for approximation quality remain, particularly for the online learning algorithm.  More work should be done to resolve these issues.  Therefore, I retain my score of a 5 (marginally below the acceptance threshold).

---

> > > ### Author Response · Authors · 2021-11-25
> > > **About guarantees of previous work**
> > >
> > > Thanks for reading the rebuttal comments and looking at the revision. Regarding approximation quality, particularly for the learning algorithm, we wanted to point out that even in the paper which introduced the low-rank decomposition for NDPPs [1], the only learning guarantee they have [Theorem 1] is about the time required for computing the log-likelihood and the gradients. The paper doesn't say anything about the approximation quality of the objective (comparing their algorithm's solution value to the global maxima). We don't think there are any such learning guarantees in the current NDPP literature. If we missed something, please do let us know. Also, regarding the approximation quality guarantees for the greedy algorithm for MAP Inference (Theorem 2) in [1], the assumptions are quite strong and don't really hold for real-world datasets (we tried to compute some of the parameters they mention since we also use them in our guarantees for the streaming algorithm). Therefore, even though [1] has some theoretical guarantees on the approximation quality of the MAP solution (and no guarantees for the learning objective), its main "effectiveness" guarantees are also primarily empirical.
> > >
> > > [1] Scalable Learning and MAP Inference for Nonsymmetric Determinantal Point Processes, Mike Gartrell, Insu Han, Elvis Dohmatob, Jennifer Gillenwater, Victor-Emmanuel Brunel, ICLR 2021.

---

> > > > ### Comment · Reviewer_9kPN · 2021-11-25
> > > > **Clarification regarding approximation guarantees for online learning algorithm**
> > > >
> > > > Thanks for the additional comments.  To clarify what I mean regarding an approximation guarantee for the online learning algorithm: here I am *not* referring to a guarantee that optimization of the non-convex NDPP objective function converges to a global maximum, which is a much more difficult problem; such guarantees are not generally available even for symmetric DPPs (except for very restricted cases).  Instead, I am referring to the fact that the approximate objective function (Eq. 4) optimized by the proposed online learning algorithm is not the same as the standard (offline or "ground-truth") NDPP objective function, as pointed out in my original review (and by reviewer J9Ht), and in your general response to all reviewers.  More theoretical understanding of the optimization of Eq. 4, and how it relates to the standard NDPP objective, should be developed.

---

### Official Review · Reviewer_NfZW · 2021-11-02

**Correctness:** 3
**Technical Novelty And Significance:** 3
**Empirical Novelty And Significance:** 3
**Recommendation:** 5
**Confidence:** 4

**Main Review:**

##  Strengths
- Clarity: this paper is well-exposed and the intuition behind the algorithms is clear (the background exposition on NDPPs is also very well written and accessible).
- Novelty: as the authors mention, this work provides the first analysis of MAP-inference and learning for the streaming settings of NDPPs.
- Empirical evaluation: the authors show that the online 2-neighbor greedy MAP algorithm improves upon _the offline algorithm_, which is a surprising and meaningful result.

## Weaknesses
- The streaming context is novel for NDPPs. However, as the authors point out, there exists previous work looking into streaming for standard DPPs [1]. Comparing the proposed algorithms to [1] to verify that the lack of symmetry provides similar benefits in the streaming setting to those in the offline setting stands out as a missing comparison.
- The first streaming algorithm (Algorithm 1) is quite simplistic, and is more interesting as a baseline rather than as a contribution in of itself.
- The core idea behind the better performing MAP algorithms (online LSS, online 2-neighbor) lies in the construction of a "stash" of discarded items. This idea was introduced in [1] for the MAP inference of (symmetric) DPPs, but the explicit connection between these two works is not made in this paper. If there are crucial differences between the stashes introduced in [1] and the stash used in this work, this should be mentioned explicitly and in detail; if the two ideas are similar, this should be discussed prominently in this work. Similarly, the results in Theorem 5 are (as far as I can tell) almost identical to those of [1, Theorem 3.1]. Again, this should be discussed explicitly in this paper. (Relatedly, I think there is a confusion between $\epsilon$ (used in [1]) and $\alpha$ (used here) to describe Online LSS, since I believe the authors use $\epsilon$ to describe Online-LSS in the experimental section.

## Questions / comments
- In [1], algorithm 1's dependency on Δ is an issue, since Δ can be arbitrarily small. Does a similar issue arise for Online LSS and Online 2-neighbor? If so, can this be addressed?
- More generally, being explicit about how the lack of symmetry in the DPP changes how the streaming setting must be approached would make for an interesting contribution of this work.
- Could 2-neighbor be extended to arbitrary sizes of subsets (3-neighbor, etc.)? Understanding at which point the degree of interactions cease to provide benefits that are worth the increase in memory/time constraints would be a valuable contribution to the DPP community: are pairwise interactions, as in 2-neighbor, sufficient to characterize most of the necessary DPP properties?
- The derivation of the gradient updates for the online algorithm can be removed from the main paper.
- Can the authors provide any insight into the stunning performance of the online learning algorithm? Compared to the offline algorithm, the online learning seems to converge almost an order of magnitude faster. It would be interesting to see if it's possible to switch from the online to the offline algorithm after the initial jump in log-likelihood, to achieve the best of both worlds (fast convergence, low NLL).
- Am I correct in understanding that Figure 1 reports the volume rather than the log volume? If so, the authors should consider log-warping the evaluation function $f$, since improvements in the range of $10^{-20}$ are difficult to gauge.
- Can you clarify the experimental conditions used for the offline learning algorithm (batch size, etc.)?

[1] Online MAP Inference of Determinantal Point Processes, Bhaskara et al., 2020

**Summary Of The Paper:**

This paper introduces MAP-inference and learning algorithms for nonsymmetric determinantal point processes (NDPPs) in the streaming and online settings.

This paper provides the first analysis of NDPP-related algorithms within a streaming context; its contributions include the algorithms themselves, theoretical analyses (algorithm guarantees, space and time complexity), and experimental evaluation of these algorithms across several standard DPP-evaluation datasets.

**Summary Of The Review:**

This paper proposes the first analysis of MAP inference and learning of nonsymmetric DPPs in a streaming setting; the authors propose novel algorithms, provide guarantees and complexity analyses, and evaluate their algorithms empirically across a variety of benchmarks. Startlingly, the authors show that their online algorithms are competitive with (and often outperform) their offline equivalents.

My main concern with this paper is novelty: there is significant overlap between the MAP-inference section of this work and previous work by Bhaskara et al. (2020), both in terms of the key ideas (using a stash) and in how the algorithms are analyzed. If this overlap is only in appearance, the authors should discuss in detail where their contributions depart from this previous work. Currently, it is difficult to understand the extent of the novelty of this work.

---

> ### Author Response · Authors · 2021-11-23
> **Response to Reviewer NfZW**
>
> Thank you for your review. We have now provided a hard instance in the appendix which separates online MAP inference on NDPPs from DPPs. This should also clear your question about dependency on Delta (yes, Delta can be arbitrarily small).
>
> We agree that the first streaming algorithm is very simple. The main reason we have chosen to include it is because it is one for which we have a provable guarantee on the approximation quality. Hopefully, some of these ideas might be useful for future proofs on approximation quality of our more complex online algorithms.
>
> The extension of 2-neighbor to subsets of arbitrary sizes is an interesting question. We leave this to future work.
>
> We have moved the derivation of gradient updates to the appendix.
>
> Yes, Figure 1 reports the quantity which is the generalization of volume in the case of DPPs. Thanks for the suggestion. Since the main goal of Figure 1 was only to compare the performance of different algorithms, we don’t think changing the det to log det would make much of a difference to our main claims. But we will try it out.
>
> For the offline learning algorithm, we used a batch size of 800 for both Instacart and MovieLens (we used the same default hyperparameters for Instacart as the paper which introduced the algorithm [1]).
>
> Please see the common response for some of your questions regarding comparisons between NDPPs and symmetric DPPs.

---

### Official Review · Reviewer_FZX8 · 2021-11-02

**Correctness:** 3
**Technical Novelty And Significance:** 3
**Empirical Novelty And Significance:** 3
**Recommendation:** 6
**Confidence:** 3

**Main Review:**

The problems that this paper studies are very interesting. Several algorithms are proposed to solve these problems, and the effectiveness of the algorithms are verified through experiments.

The technical sections are a bit hard to follow, and a lot of details are omitted to save space. For example, for outline of Algorithm 2, the auxiliary set $T$ is defined but its size estimate is not given. It would be great to include some intuition in the main paper.

**Summary Of The Paper:**

This paper introduces the streaming and online MAP inference and learning problems for NDPPs.
- For streaming MAP inference, an algorithm is proposed with total time linear in $n$ and memory that is constant in $n$;
- For online MAP inference, several algorithms are proposed such that at any point in time a valid solution is maintained;
- For online learning algorithm, a single pass algorithm is proposed with memory that is constant in $m$;
- Experiments are conducted to show that these streaming and online algorithms achieves comparable performance to state-of-the-art offline algorithms.

**Summary Of The Review:**

Overall I think the problems that this paper studies are interesting and the proposed algorithm are effective.

---

> ### Author Response · Authors · 2021-11-23
> **Response to Reviewer FZX8**
>
> Thank you for your review. We have provided an upper bound on the size of the stash in the appendix but could not do so in the main paper due to space constraints. Please see the common response about intuition for the algorithms.

---

### Official Review · Reviewer_J9Ht · 2021-11-03

**Correctness:** 2
**Technical Novelty And Significance:** 1
**Empirical Novelty And Significance:** 2
**Recommendation:** 3
**Confidence:** 5

**Main Review:**

Strengths:
- The paper studies a new problem of NDPPs under the online setting that has not been covered before.
- The authors properly apply the prior online algorithm for greedy submodular maximization to the NDPPs
- Experimental results are convincing that the effectiveness of the proposed online inference and learning algorithms for NDPPs

Weaknesses:
- The writing quality needs to be improved. The manuscript contains several typos, notational abusements which are provided with minor comments (below). Also, the paper simply introduces the proposed algorithms without any justification or intuition, which is hard to understand how the authors deal with problems under the online settings.
- Moreover, some algorithms are already proposed in prior works, but there is no reference. For example, Algorithm 2 (Online-LSS) was proposed in [1].

  [1] Bhaskara et al., Online MAP Inference of Determinantal Point Processes, NeurIPS, 2019
- It is not clear how the streaming setting in section 4 is different from the online setting in section 5. Since both Algorithm 1 and 4 take sequential inputs (with random order or on-the-fly), it seems that both can be used for both settings. It would be great if more detailed descriptions of streaming and online settings are provided.
- In section 5, the authors provide 2 different algorithms, i.e., local search and 2-neighborhood local search. Comparing Theorem 5 and 7, the latter has no gain in terms of runtime complexity. However, it shows better empirical performance. What is the reason for it? Can the optimality of these algorithms be analyzed?
- The authors propose the approximate objective (Eq (4)) for learning NDPPs under the online setting. However, this is somewhat very different from the log-likelihood of DPP because the objective contains $\log \det(L_S) -\log \det(L_S + I_S)$. Does learning this objective guarantee convergence of the ground-truth NDPP objective? How does the approximated objective Eq (4) relate to the offline version of the MLE objective?

Minor comments:

  - A comma is missing in the fourth line of the second paragraph on page 1.
  - Please edit “state-of-the-art” in the second paragraph on page 2.
  - Please edit “are minimize” -> “are minimizing” in the second last row on page 3.
  - It would be good to place Algorithm 4 in the main manuscript.
  - In Theorem 5 and 7, please edit $f(S) = \det( V_S^\top V_S + B_S^\top C B_S)$
  - What is $j_{\mathrm{\max}}$ in line 9 in Algorithm 1?


**Summary Of The Paper:**

This paper studies the online inference and learning problems for nonsymmetric determinantal point processes (NDPPs). The authors use the online greedy algorithm for MAP inference and modify the learning objective for being suitable in the online setting. Experiments with real-world datasets show that the proposed online algorithms are comparable or even better than state-of-the-art offline algorithms.

**Summary Of The Review:**

Although this paper firstly studies new problems of online NDPPs, it has a lack of algorithm novelty, theoretical analyses as well as writing quality for addressing their methodology. Hence, the paper should be improved for acceptance.

---

> ### Author Response · Authors · 2021-11-23
> **Response to Reviewer J9Ht**
>
> Thank you for your review. We have corrected all the typos you have mentioned. We have also placed Algorithm 4 in the main manuscript. The streaming algorithm we have proposed doesn’t work in the online setting because it only gives a valid solution at the end of the stream (and it needs to know the length of the stream before it begins it’s run). For your questions regarding optimality of the MAP Inference algorithms and the approximate learning objective, please see the common response.

---

### Author Response · Authors · 2021-11-23
**General response to all reviewers**

First of all, we would like to thank all the reviewers for their valuable feedback. We will first address some comments which were common across multiple reviews and will respond to individual reviews separately. We have updated the draft taking a good chunk of the reviewer feedback into account. Please have a look at the updated version.

We want to stress that our paper gives the first problem formulations for streaming and online versions of MAP Inference of low-rank-NDPPs (this problem formulation is not-at-all obvious from any of the previous works. We spent a non-trivial amount of time coming up with this formulation). Our formulations themselves give a good structure on how to store extremely large NDPP models (models which cannot fit by themselves in RAM) i.e. store the C matrix separately and all the v_i and b_i as (v_i, b_i) pairs (the straight-forward way to store the data would be to store V,C,B separately (as in the open-source code provided by the Scalable NDPPs ICLR 2021 paper [1]).

Once we do have these formulations, yes, our Online-LSS algorithm strictly generalizes the Online-LS algorithm from the Online-DPPs paper [2]. We did cite the paper in the related works section but we have now cited it again in the Online-LSS section of the paper. We want to stress again that this generalization is (sort of straight-forward) only because our online problem formulation generalizes the online problem formulation of [2].

Regarding justification and intuition for MAP Inference algorithms: We agree that having a clean explanation for why our MAP inference algorithms work would be nice to have. Unfortunately, NDPPs are significantly more general (and complex) than DPPs and unlike the case for DPPs where the objective function corresponds to a nice geometric notion i.e. volume, the objective function for MAP Inference on NDPPs doesn’t have a corresponding clean notion (if any of the reviewers are aware of such notions, please feel free to share it with us). This is actually a core issue because of which it is not at all clear how the proofs of approximation factors for similar algorithms for DPPs would generalize to NDPPs (the proof techniques for DPPs from [2] for Online-LS heavily use the fact that the objective function is a volume and thus use some coresets developed for geometric problems). Actually, we believe it might be impossible to prove any bounded approximation factor guarantees for our algorithms without any additional assumptions. We have provided a hard instance in the appendix on which all of our algorithms fail. But note that this certainly doesn’t mean that our algorithms aren’t very useful in practice (as our experiments demonstrate). The hope is that some theorist might be inspired by our paper and prove guarantees for our online algorithms under reasonable assumptions on the data in future work.

Regarding the learning objective (Eq 4), we do not make any claim in the paper that optimizing this objective leads to any convergence of the ground-truth NDPP objective. That would be wonderful to have but unfortunately, we don’t know whether we can show any such statement. Our learning section’s main contributions are to shed light on the online learning problem for NDPPs and our algorithm (or heuristic, if you wish to call it) is only to show that a simple online algorithm can give good empirical performance.

Some insight regarding the speedup for the learning: Effectively, we are doing 2 approximations at every time step. One is that we only use the subset seen at that time step. Two is that we only update the representation of items in that particular subset, not the whole set of items. If we think of all the items as vertices in a graph, with edges between items if they appear together in subsets, then at every time step, instead of propagating the changes from one subset to the entire graph, we are restricting it to only that subset. In practice, one reason why we get fast convergence might be that the weight of interactions between items might be very small but the offline algorithm spends a lot of resources to update all the item representations in every time step.

References:

[1]: Scalable Learning and MAP Inference for Nonsymmetric Determinantal Point Processes: Mike Gartrell, Insu Han, Elvis Dohmatob, Jennifer Gillenwater, and Victor-Emmanuel Brunel, ICLR 2021.

[2]: Online MAP Inference of Determinantal Point Processes: Aditya Bhaskara, Amin Karbasi, Silvio Lattanzi, and Morteza Zadimoghaddam, NeurIPS 2020)

---

### Public Comment · ~Aravind_Reddy1 · 2022-05-20
**Thanks again for the feedback! Revised version accepted to ICML 2022 :)**

We are happy to let you know that a revised version of this paper titled "One-Pass Algorithms for MAP Inference of Nonsymmetric Determinantal Point Processes", in which we focus exclusively on the MAP Inference portion of this submission, was recently accepted to ICML 2022 as a short presentation. The reviews provided by the ICLR reviewers were very beneficial to us in revising our paper. We wanted to thank all of you once again!

---

### Decision · Program_Chairs · 2022-01-20

**Decision:**

Reject

**Comment:**

This paper studies online MAP inference and learning for nonsymmetric determinantal point processes (NDPPs). The main contribution is an online greedy algorithm. Surprisingly they show that their algorithm outperforms various offline algorithms on real-world datasets. That said, the main concern was the novelty with respect to the prior work of Bhaskara et al. who gave an online approximation algorithm for MAP inference in DPPs. To compare the two works: (1) Bhaskara et al. give an algorithm for DPPs, and NDPPs are more complex (2) Bhaskara et al. give provable guarantees on the approximation ratio, but no such guarantees are known for NDPPs (3) And finally, some of the key ingredients in the online algorithm for NDPPs, like the stash, were already in the work of Bhaskara et al. Overall the reviewers felt that this submission would be improved with a clearer discussion of the contributions over prior work.